# Stochastic Frank Wolfe
# for Constrained Nonconvex Optimization

## Abstract

We provide a practical convergence analyses of Stochastic Frank Wolfe (SFW) and SFW with momentum with constant and decaying learning rates for constrained nonconvex optimization problems. We show that a convergence measure called the Frank Wolfe gap converges to zero only when we decrease the learning rate and increase the batch size. We apply FW algorithms to adversarial attacks and propose a new adversarial attack method, Auto-FW. Finally, we compare existing methods with the FW algorithms in attacks against the latest robust models.

## 1 Introduction

### 1.1 Background

Nonconvex optimization is necessary for training deep neural networks. First-order methods, such as stochastic gradient descent (SGD) (Robbins & Monro, 1951), adaptive moment estimation (Adam) (Kingma & Ba, 2015) and their variants (Polyak, 1964; Nesterov, 1983; Duchi et al., 2011; Tieleman & Hinton, 2012; Reddi et al., 2018), are still very powerful methods, and their convergence analysis for nonconvex optimization has been widely studied (Fehrman et al., 2020; Bottou et al., 2018a; Scaman & Malherbe, 2020; Loizou et al., 2021; Zaheer et al., 2018; Zou et al., 2019; Chen et al., 2019; Zhou et al., 2020; Chen et al., 2021; Iiduka, 2022).

When solving constrained optimization problems with these methods, it is necessary to compute the projection onto the constraint set at each iteration. In many problem settings, the computational cost of projections to the constraint set such as the Euclidean norm ball can be very high, and in extreme cases the projections can even be computationally infeasible (Collins et al., 2008). Here, we focus on the Frank Wolfe algorithm (Frank & Wolfe, 1956), also called the conditional gradient algorithm (Levitin & Polyak, 1966), a projection-free first-order method for constrained optimization.

The Frank Wolfe algorithm is a classical first-order method for solving convex optimization problems with compact convex constraint sets. In recent years, it has received renewed attention thanks to its ability to efficiently handle structured constraints that appear in machine learning. The algorithm and its many variants, such as stochastic Frank Wolfe (SFW), have been well studied in the convex or strongly convex setting (Jaggi, 2013; Freund & Grigas, 2014; Lacoste-Julien & Jaggi, 2015; Goldfarb et al., 2017; Locatello et al., 2017; Zhang et al., 2019c; Tang et al., 2022), and they have been applied to matrix completion (Freund et al., 2017; Locatello et al., 2019), regression (Négiar et al., 2020; Dvurechensky et al., 2023; Wirth et al., 2023), and support vector machine (SVM)(Hazan & Luo, 2016; Lu & Freund, 2021). Even in the nonconvex setting, convergence analyses have been provided for many variants (Reddi et al., 2016; Gu et al., 2019; Grigas et al., 2019; Yurtsever et al., 2019; Chen et al., 2020a; Pokutta et al., 2020; Sahu & Kar, 2020; Combettes et al., 2021; Nazykov et al., 2024), and some have been successful in experiments on deep neural networks (DNNs) (Berrada et al., 2019; Miao et al., 2022). In particular, Frank Wolfe-type algorithms have been shown to be effective in making adversarial attacks (see Section 4).

## 1.2 Motivation

**1. Weak convergence analysis.** Several previous studies provide convergence analyses of SFW methods for nonconvex optimization, but many analyses do not actually show that some convergence measure tends to $0$ as the number of steps $T \to \infty$. Some of the previous studies present guarantees of convergence by deriving the inequality, $\frac{1}{T} \sum_{t=0}^{T-1} \textbf{gap} \leq \frac{A}{T\gamma} + B\gamma$, and use $\gamma = \frac{1}{\sqrt{T}}$ to derive $\frac{1}{T} \sum_{t=0}^{T-1} \textbf{gap} \leq \frac{A}{\sqrt{T}} + \frac{B}{\sqrt{T}} = \mathcal{O}\left(\frac{1}{\sqrt{T}}\right)$, where **gap** is the convergence criterion, $T$ the number of steps, $\gamma > 0$ the learning rate, and $A, B > 0$ constants for simplicity. Recall that convergence of a sequence $(a_n)$ to $0$ is a necessary and sufficient condition of the following: $\forall \epsilon > 0, \exists n_0 \in \mathbb{N} \colon \forall n \geq n_0 \Rightarrow |a_n| < \epsilon$. Therefore, if the learning rate is set as $\gamma = \frac{1}{\sqrt{T}}$, the total number of iterations $T$ is predetermined and fixed, so $\frac{A}{\sqrt{T}} + \frac{B}{\sqrt{T}}$ is a constant, $T$ cannot approach infinity, and **gap** is not guaranteed to converge to $0$. This is evident from the fact that $\frac{1}{T} \sum_{t=0}^{T-1} \textbf{gap} \leq \frac{A}{T\gamma} + B\gamma$ for any $T$, and even if $T \to \infty$, convergence of **gap** to $0$ is still not guaranteed due to the extra term $B\gamma$. Thus, these analyses do not exactly show $\frac{1}{T} \sum_{t=0}^{T-1} \textbf{gap} = \mathcal{O}\left(\frac{1}{\sqrt{T}}\right)$ and $\frac{1}{T} \sum_{t=0}^{T-1} \textbf{gap} \to 0 \; (T \to \infty)$. Instead, from $\frac{A+B}{\sqrt{T}} < \epsilon$ i.e. $T > \frac{(A+B)^2}{\epsilon^2}$, one can figure out the behavior of the number of iterations $T$ for a fixed threshold $\epsilon$. Thus, these convergence analysis focuses on how many iterations $T$ are required for a certain fixed threshold $\epsilon$, and their convergence rate of $\mathcal{O}\left(\frac{1}{\sqrt{T}}\right)$ does not necessarily mean that **gap** converges to $0$ at rate of $\mathcal{O}\left(\frac{1}{\sqrt{T}}\right)$. Therefore, the purpose of this paper is to perform a convergence analysis of SFW such that **gap** converges to $0$ based on the definition of convergence of a sequence.

Table 1: Summary of previous studies of SFW methods in nonconvex optimization. $T$ means the total number of iterations and $t \in [T]$ denotes an iteration or time. $L$ means Lipschitz constant. $\gamma_t = \mathcal{O}\left(\sqrt{\frac{1}{TL}}\right)$ in Learning Rate column indicates that the learning rate $\gamma_t$ is set at $\gamma_t = \frac{K}{TL}$ using some positive constant $K$, emphasizing in particular that it is based on $T$ and $L$. "Noise" in the Batch Size column means that algorithm uses noisy observation, i.e., $g(\boldsymbol{\theta}) = \nabla f(\boldsymbol{\theta}) + (\text{Noise})$, of the full gradient $\nabla f(\boldsymbol{\theta})$, while $b = X$ in the Batch Size column means that algorithm uses a mini-batch gradient $\nabla f_{\mathcal{S}_t}(\boldsymbol{\theta}) = \frac{1}{b} \sum_{i \in [b]} \mathsf{G}_{\xi_{t,i}}$ with a batch size $b \, (\leq n)$, where $\mathsf{G}_{\xi_{t,i}}$ is stochastic gradient and $n$ is the number of training data. The Momentum column states whether the SFW algorithm includes a momentum factor. These results were presented in (1)(Reddi et al., 2016, Theorem 2), (2)(Gu et al., 2019, Theorem 1), (3)(Grigas et al., 2019, Theorem 2.1), (4)(Négiar et al., 2020, Theorem 2), (5)(Combettes et al., 2021, Theorem 3.3), and (6)(Nazykov et al., 2024, Theorem 2.3), where $\mathcal{G}(\boldsymbol{\theta})$ is Frank Wolfe gap (see Section 2.1) and $D > 0$ means diameter of the constraint set (see Assumption (A1)).

| Algorithm | Learning Rate | Batch Size | Momentum | Convergence Analysis |
|---|---|---|---|---|
| (1) SFW | $\gamma_t = \mathcal{O}\left(\sqrt{\frac{1}{TL}}\right)$ | $b = T$ | No | $\frac{1}{T} \sum_{t \in [T]} \mathbb{E}[\mathcal{G}(\boldsymbol{\theta}_t)] = \mathcal{O}\left(\frac{1}{\sqrt{T}}\right)$ |
| (2) AsySFW | $\gamma_t = \mathcal{O}\left(\sqrt{\frac{1}{TL}}\right)$ | $b = T$ | No | $\frac{1}{T} \sum_{t \in [T]} \mathbb{E}[\mathcal{G}(\boldsymbol{\theta}_t)] = \mathcal{O}\left(\frac{1}{\sqrt{T}}\right)$ |
| (3) FW-SD | $\gamma_t = \mathcal{O}\left(\sqrt{\frac{1}{L}}\right)$ | $b = T$ | No | $\mathbb{E}\left[\mathcal{G}(\boldsymbol{\theta}_T)\right] = \mathcal{O}\left(\frac{1}{\sqrt{T}}\right)$ |
| | $\gamma_t = \mathcal{O}\left(\sqrt{\frac{1}{L}}\right)$ | $b = t$ | No | $\mathbb{E}\left[\mathcal{G}(\boldsymbol{\theta}_T)\right] = \mathcal{O}\left(\sqrt{\frac{\log(T)}{T}}\right)$ |
| (4) SFW | $\gamma_t = \frac{2}{t+2}$ | $b \leq n$ | No | $\liminf_{t \to \infty} \mathbb{E}[\mathcal{G}(\boldsymbol{\theta}_t)] = 0$ |
| (5) AdaSFW | $\gamma_t = \sqrt{\frac{1}{T}}$ | $b_t = \mathcal{O}\left(\frac{T}{L}\right)$ | No | $\frac{1}{T} \sum_{t \in [T]} \mathbb{E}[\mathcal{G}(\boldsymbol{\theta}_t)] = \mathcal{O}\left(\frac{1}{\sqrt{T}}\right)$ |
| | $\gamma_t = \frac{1}{t+1}$ | $b_t = \mathcal{O}\left(\frac{t}{L^2}\right)$ | No | $\limsup_{t \to \infty} \mathbb{E}[\mathcal{G}(\boldsymbol{\theta}_t)] \leq 0$ |
| (6) any SFW methods | $\gamma_t = \sqrt{\frac{1}{T}}$ | Noise | No | $\min_{t \in [T]} \mathbb{E}\left[\mathcal{G}(\boldsymbol{\theta}_t)\right] = \mathcal{O}\left(\frac{1}{\sqrt{T}} + D\right)$ |

Table 1 summarizes previous studies on SFW methods for nonconvex optimization. Some of the previous studies on Frank Wolfe methods (Reddi et al., 2016; Gu et al., 2019; Chen et al., 2020a; Mokhtari et al., 2020; Nazykov et al., 2024) used a learning rate $\gamma$ that includes the total number of iterations $T$ such that $\gamma = \frac{1}{\sqrt{T}}$. Even if $T$ is predetermined, this setup may be experimentally unrealistic, since $T$ can be very large depending on the training dataset and the number of epochs. Reddi et al. (2016); Gu et al. (2019) used a batch size determined by $b = T$, but this setting is not realistic from the standpoint of computational complexity because $b$ becomes too large for medium-sized or larger experiments. Some studies (Reddi et al., 2016; Gu et al., 2019; Grigas et al., 2019) also include a Lipschitz constant in the learning rate. Since that information is not available in advance, this setup would also be impractical. A few studies (Négiar et al., 2020; Combettes et al., 2021) used a traditional learning rate that decreases as $\gamma_t := \frac{2}{t+2}$ depending on time $t$. This setting may not be suitable for practical use, especially for large-scale optimizations such as DNN training, because the learning rate quickly becomes too small. These analyses do not explain the effectiveness of SFW methods in large-scale optimization of DNNs as is evident in (Miao et al., 2022). We therefore aim to provide an analysis of the convergence of SFW when using constant and decaying learning rates, which would be experimentally realistic. We also aim to provide a similar analysis for SFW with momentum (SFWM), a natural extension of SFW.

**2. Are FW attacks effective against robust models?** Chen et al. (2020a) showed that the Frank Wolfe algorithm is effective in adversarial attacks against non-robust models. Therefore, we would like to clarify whether it is effective against robust models. Furthermore, we propose Auto-FW (AFW), inspired by Auto-Projected Gradient Descent (APGD) (Croce & Hein, 2020) and Auto-Conjugate Gradient Descent (ACG) (Yamamura et al., 2022), and clarify its performance for robust models (see Section 4 for details).

### 1.3 Contribution

**1. Practical convergence analysis of SFW and SFW with momentum (Section 3).** We provide convergence analyses of SFW (Algorithm 2) and SFW with momentum (Algorithms 3) using a user-defined learning rate that is independent of unknowns that cannot be known a priori. To evaluate convergence, we use the Frank Wolfe gap $\mathcal{G}(\boldsymbol{\theta})$, which is a commonly used measure in convergence analyses of Frank Wolfe algorithms (see Section 2.1 for details).

Let $(\boldsymbol{\theta}_t) \in \mathbb{R}^d$ be the sequence generated by each of SFW (Algorithm 2) and SFWM (Algorithm 3). In Section 3, we will show that, under certain assumptions, the average of $\mathbb{E}[\mathcal{G}(\boldsymbol{\theta}_t)]$ has an upper bound as shown in Table 2, where the constant learning rate is $\gamma_t := \gamma$ and the constant batch size is $b_t = b$. In addition, for fixed natural numbers $K$ and $E$, the decaying learning rate and increasing batch size are defined as follows:

Decaying Learning Rate (I): $\gamma_t := \dfrac{1}{t+1}$,

Decaying Learning Rate (II): $\gamma_t := \dfrac{1}{(t+1)^a}$ $(a \in [0.5, 1))$,

Decaying Learning Rate (III): $\gamma_t := (\underbrace{\gamma, \gamma, \cdots, \gamma}_{K}, \underbrace{\eta\gamma, \eta\gamma, \cdots, \eta\gamma}_{K}, \cdots, \underbrace{\eta^{P-1}\gamma, \eta^{P-1}\gamma, \cdots, \eta^{P-1}\gamma}_{K})$, (1)

Increasing Batch Size: $b_t := (\underbrace{b, b, \cdots, b}_{E}, \underbrace{\lambda b, \lambda b, \cdots, \lambda b}_{E}, \cdots, \underbrace{\lambda^{Q-1}b, \lambda^{Q-1}b, \cdots, \lambda^{Q-1}b}_{E})$, (2)

where $\eta \in (0,1)$, $\lambda > 1$, $PK = T$, and $QE = T$. $\gamma > 0$ is the initial learning rate and $b$ $(\leq n)$ is the initial batch size, where $n$ is the number of training data. In addition, in Decaying Learning Rate (III), we set a lower bound $\underline{\gamma} > 0$, and if $\gamma_t$ computed according to the definition (1) is less than the lower bound $\underline{\gamma}$ $(\leq \gamma_t)$, we set $\gamma_t = \underline{\gamma}$. Similarly, in the increasing batch size, we set $b_t := n$ if $b_t$ computed according to the definition (2) is above the upper bound $n$.

Note that all of our theorems are common to SFW and SFWM and that the momentum factor does not appear. This is due to our key lemma (Lemma A.3) and is one of our technical contributions. Table 2 shows that the extra term independent of $T$ disappears from the upper bound of gap only when using

Table 2: Convergence rate of our analysis. (LR: learning rate). Note that $C > 0$ is a constant.

| | Constant Batch Size | Increasing Batch Size |
|---|---|---|
| Constant LR | $\mathcal{O}\left(\dfrac{1}{T} + \dfrac{1}{\sqrt{b}} + \gamma\right)$ (Theorem 3.1) | $\mathcal{O}\left(\dfrac{1}{T} + \gamma\right)$ (Theorem 3.2) |
| Decaying LR (I) | $\mathcal{O}\left(\dfrac{\log T}{T} + \dfrac{1}{\sqrt{b}} + C\right)$ (Theorem 3.3(i)) | $\mathcal{O}\left(\dfrac{\log T}{T} + C\right)$ (Theorem 3.4(i)) |
| Decaying LR (II) | $\mathcal{O}\left(\dfrac{1}{T^{\min\{1-a,a\}}} + \dfrac{1}{\sqrt{b}}\right)$ (Theorerm 3.3(ii)) | $\mathcal{O}\left(\dfrac{1}{T^{\min\{1-a,a\}}}\right)$ (Theorem 3.4(ii)) |
| Decaying LR (III) | $\mathcal{O}\left(\dfrac{1}{T} + \dfrac{1}{\sqrt{b}}\right)$ (Theorem 3.3(iii)) | $\mathcal{O}\left(\dfrac{1}{T}\right)$ (Theorem 3.4(iii)) |

both decreasing learning rates (II) and (III) and an increasing batch size, resulting in $\frac{1}{T}\sum_{t=0}^{T-1}\mathbb{E}\left[\mathcal{G}(\boldsymbol{\theta}_t)\right] \to 0$ ($T \to \infty$). In particular, SFW and SFWM have a convergence rate of $\mathcal{O}(1/T)$ when using a decaying learning rate (III) and an increasing batch size. We applied these algorithms to deep-learning training to verify their performance (Section 3.3). Note that SFW with increasing batch sizes has been studied by (Goldfarb et al., 2017; Hazan & Luo, 2016; Reddi et al., 2016) and SGD with increasing batch size also has been well studied by (Byrd et al., 2012; Friedlander & Schmidt, 2012; Balles et al., 2017; De et al., 2017; Bottou et al., 2018b; Smith et al., 2018).

**2. Application to adversarial attack (Section 4).** Our convergence analysis can be applied to any constrained nonconvex optimization problem. In this paper, the SFW algorithms are used to generate adversarial examples. In Section 4.3, we propose a new adversarial attack method, Auto-FW (AFW), an adaptation of the APGD approach to SFW. Furthermore, we show that AFW has comparable performance to APGD, which itself has state-of-the-art performance, and discuss its limitation in Section 4.5.

## 2  Preliminaries

Let $\mathbb{N}$ be the set of non-negative integers. For $m \in \mathbb{N} \setminus \{0\}$, define $[m] := \{1, 2, \ldots, m\}$. Let $\mathbb{R}^d$ be a $d$-dimensional Euclidean space with inner product $\langle \cdot, \cdot \rangle$, which induces the norm $\|\cdot\|$. The DNNs is parameterized by a vector $\boldsymbol{x} \in \mathbb{R}^d$, which is optimized by minimizing the empirical loss function $f(\boldsymbol{\theta}) := \frac{1}{n}\sum_{i\in[n]} f_i(\boldsymbol{\theta})$, where $f_i(\boldsymbol{\theta})$ is the loss function for $\boldsymbol{\theta} \in \mathbb{R}^d$ and the $i$-th training data $(\boldsymbol{x}_i, \boldsymbol{y}_i)$ ($i \in [n]$). $(\boldsymbol{\theta}_t)_{t\in\mathbb{N}}$, or simply $(\boldsymbol{\theta}_t)$, represents the points sequence $(\boldsymbol{\theta}_0, \boldsymbol{\theta}_1, \cdots)$. Let $\xi$ be a random variable that does not depend on $\boldsymbol{\theta} \in \mathbb{R}^d$, and let $\mathbb{E}_\xi[X]$ denote the expectation with respect to $\xi$ of a random variable $X$. $\xi_{t,i}$ is a random variable generated from the $i$-th sampling at time $t$, and $\boldsymbol{\xi}_t := (\xi_{t,1}, \xi_{t,2}, \ldots, \xi_{t,b})$ is independent of $(\boldsymbol{\theta}_k)_{k=0}^t := (\boldsymbol{\theta}_0, \boldsymbol{\theta}_1, \ldots, \boldsymbol{\theta}_t) \subset \mathbb{R}^d$, where $b$ ($\leq n$) is the batch size. The independence of $\boldsymbol{\xi}_0, \boldsymbol{\xi}_1, \ldots$ allows us to define the total expectation $\mathbb{E}$ as $\mathbb{E} = \mathbb{E}_{\boldsymbol{\xi}_0}\mathbb{E}_{\boldsymbol{\xi}_1}\cdots\mathbb{E}_{\boldsymbol{\xi}_t}$. Let $\mathsf{G}_{\boldsymbol{\xi}_t}(\boldsymbol{\theta})$ be the stochastic gradient of $f(\cdot)$ at $\boldsymbol{\theta} \in \mathbb{R}^d$. The mini-batch $\mathcal{S}_t$ consists of $b_t$ samples at time $t$, and the mini-batch stochastic gradient of $f(\boldsymbol{\theta}_t)$ for $\mathcal{S}_t$ is defined as $\nabla f_{\mathcal{S}_t}(\boldsymbol{\theta}_t) := \frac{1}{b_t}\sum_{i\in[b_t]} \mathsf{G}_{\xi_{t,i}}(\boldsymbol{\theta}_t)$.

We will impose the following conditions, which are standard ones for nonconvex optimization in deep neural networks (see, e.g., (Chen et al., 2019)).

**Assumption 2.1.**

(A1) *The domain $\Omega \subset \mathbb{R}^d$ is convex and compact with diameter $D$ such that $\forall \boldsymbol{x}, \boldsymbol{y} \in \Omega \colon \|\boldsymbol{x} - \boldsymbol{y}\| \leq D$.*

(A2) *$f_i \colon \mathbb{R}^d \to \mathbb{R}$ ($i \in [n]$) are continuously differentiable and $L$-smooth on $\Omega$, i.e.,*

$$\forall \boldsymbol{x}, \boldsymbol{y} \in \Omega : \|\nabla f_i(\boldsymbol{x}) - \nabla f_i(\boldsymbol{y})\| \leq L\|\boldsymbol{x} - \boldsymbol{y}\|.$$

(A3) *Let $(\boldsymbol{\theta}_t)_{t\in\mathbb{N}} \subset \mathbb{R}^d$ be the sequence generated by an optimizer.*
(i) *For each iteration $t$, $\mathbb{E}_{\xi_t}\left[\mathsf{G}_{\xi_t}(\boldsymbol{\theta}_t)\right] = \nabla f(\boldsymbol{\theta}_t)$.*

(ii) *There exists a nonnegative constant $\sigma^2$ such that $\mathbb{E}_{\xi_t}\left[\|\mathsf{G}_{\xi_t}(\boldsymbol{\theta}_t) - \nabla f(\boldsymbol{\theta}_t)\|^2\right] \leq \sigma^2$.*

(A4) *For each iteration $t$, the optimizer samples the mini-batch $\mathcal{S}_t \subset \mathcal{S}$ and estimates the full gradient $\nabla f$ as*

$$\nabla f_{\mathcal{S}_t}(\boldsymbol{\theta}_t) := \frac{1}{b_t} \sum_{i \in [b]} \mathsf{G}_{\xi_{t,i}}(\boldsymbol{\theta}_t) = \frac{1}{b_t} \sum_{\{i \colon (\boldsymbol{x}_i, \boldsymbol{y}_i) \in \mathcal{S}_t\}} \nabla f_i(\boldsymbol{\theta}_t).$$

**Problem 2.1.** *Under Assumption 2.1, we would like to minimize $f(\boldsymbol{\theta}) := \frac{1}{n} \sum_{i=1}^{n} f_i(\boldsymbol{\theta})$ over $\Omega$.*

## 2.1 Stochastic Frank Wolfe algorithm

---

**Algorithm 1** Frank Wolfe

**Require:** $(\gamma_t)_{t \in \mathbb{N}} \subset \mathbb{R}_{++}$
  $t \leftarrow 0, \boldsymbol{\theta}_0 \in \Omega$
  **loop**
    $\boldsymbol{v}_t = \underset{\boldsymbol{v} \in \Omega}{\mathrm{argmax}} \langle \boldsymbol{v}, -\nabla f(\boldsymbol{\theta}_t) \rangle$
    $\boldsymbol{d}_t = \boldsymbol{v}_t - \boldsymbol{\theta}_t$
    $\boldsymbol{\theta}_{t+1} = \boldsymbol{\theta}_t + \gamma_t \boldsymbol{d}_t$
    $t \leftarrow t + 1$
  **end loop**

---

**Algorithm 2** Stochastic Frank Wolfe (SFW)

**Require:** $(\gamma_t)_{t \in \mathbb{N}} \subset \mathbb{R}_{++}$
  $t \leftarrow 0, \boldsymbol{\theta}_0 \in \Omega$
  **loop**
    $\boldsymbol{v}_t = \underset{\boldsymbol{v} \in \Omega}{\mathrm{argmax}} \langle \boldsymbol{v}, -\nabla f_{\mathcal{S}_t}(\boldsymbol{\theta}_t) \rangle$
    $\boldsymbol{d}_t = \boldsymbol{v}_t - \boldsymbol{\theta}_t$
    $\boldsymbol{\theta}_{t+1} = \boldsymbol{\theta}_t + \gamma_t \boldsymbol{d}_t$
    $t \leftarrow t + 1$
  **end loop**

---

In general, $\|\nabla f(\boldsymbol{\theta})\|$ cannot be used as a measure of convergence in constrained optimization. So instead, the Frank Wolfe gap $\mathcal{G}(\boldsymbol{\theta})$ such that

$$\mathcal{G}(\boldsymbol{\theta}) := \max_{\boldsymbol{v} \in \Omega} \langle \boldsymbol{v} - \boldsymbol{\theta}, -\nabla f(\boldsymbol{\theta}) \rangle,$$

is introduced as a measure of convergence. Suppose that $\boldsymbol{\theta}^\star$ is a local minimizer of $f$ over $\Omega$; from the optimality conditions,

$$\forall \boldsymbol{v} \in \Omega : \langle \boldsymbol{v} - \boldsymbol{\theta}^\star, \nabla f(\boldsymbol{\theta}^\star) \rangle \geq 0 \ \text{ i.e. } \mathcal{G}(\boldsymbol{\theta}^\star) \leq 0.$$

Therefore, our goal is to make $\mathcal{G}(\boldsymbol{\theta})$ small. Note that $\mathcal{G}(\boldsymbol{\theta}) = 0$ does not necessarily imply that $\boldsymbol{\theta}$ is a local minimizer of $f$. We will consider the following optimizer that uses the mini-batch stochastic gradient $\nabla f_{\mathcal{S}_t}(\boldsymbol{\theta}_t)$ instead of the full gradient $\nabla f(\boldsymbol{\theta}_t)$, described by Algorithm 2, for solving Problem 2.1. In addition, attempts have been made to add a momentum term in several different variants and under many different names (Mokhtari et al., 2018; Chen et al., 2018; Mokhtari et al., 2020; Xie et al., 2020; Chen et al., 2020a; Pokutta et al., 2020; Pethick et al., 2025). In accordance with Pokutta et al. (2020), we will refer to Algorithm 3 as SFW with momentum (SFWM). Note that our SFWM have fixed momentum factor $\beta$.

---

**Algorithm 3** SFW with momentum (SFWM)

**Require:** $(\gamma_t)_{t \in \mathbb{N}} \subset \mathbb{R}_{++}$, $\beta \in [0, 1)$
  $t \leftarrow 0, \boldsymbol{\theta}_0 \in \Omega, \boldsymbol{m}_{-1} \leftarrow \boldsymbol{0}$
  **loop**
    $\boldsymbol{m}_t = \beta \boldsymbol{m}_{t-1} + (1 - \beta) \nabla f_{\mathcal{S}_t}(\boldsymbol{\theta}_t)$
    $\boldsymbol{v}_t = \underset{\boldsymbol{v} \in \Omega}{\mathrm{argmax}} \langle \boldsymbol{v}, -\boldsymbol{m}_t \rangle$
    $\boldsymbol{d}_t = \boldsymbol{v}_t - \boldsymbol{\theta}_t$
    $\boldsymbol{\theta}_{t+1} = \boldsymbol{\theta}_t + \gamma_t \boldsymbol{d}_t$
    $t \leftarrow t + 1$
  **end loop**

---

# 3 Theoretical Main Results

## 3.1 Convergence analyses of SFW algorithms using a constant learning rate

**Assumption 3.1.** (C1) $\gamma_t := \gamma$ for all $t \in \mathbb{N}$.

The following are convergence analyses of Algorithm 3 with a constant learning rate. The proofs of Theorems 3.1 and 3.2 are given in Appendix B.

**Theorem 3.1** (**Constant Learning Rate and Constant Batch Size**). *Suppose that Assumptions 2.1 and 3.1 hold and consider the sequence $(\boldsymbol{\theta}_t)_{t\in\mathbb{N}}$ generated by each of Algorithms 2 and 3 with a constant batch size $b_t := b$. Then, the following holds:*

$$\frac{1}{T}\sum_{t=0}^{T-1}\mathbb{E}\left[\mathcal{G}(\boldsymbol{\theta}_t)\right] \leq \frac{f(\boldsymbol{\theta}_0) - f(\boldsymbol{\theta}^\star)}{\gamma T} + \frac{D\sigma}{\sqrt{b}} + \frac{LD^2\gamma}{2} = \mathcal{O}\left(\frac{1}{T} + \frac{1}{\sqrt{b}} + \gamma\right),$$

*where $\boldsymbol{\theta}^\star := \underset{\boldsymbol{\theta}\in\Omega}{\operatorname{argmin}} \ f(\boldsymbol{\theta})$.*

**Theorem 3.2** (**Constant Learning Rate and Increasing Batch Size**). *Suppose that Assumptions 2.1 and 3.1 hold and consider the sequence $(\boldsymbol{\theta}_t)_{t\in\mathbb{N}}$ generated by each of Algorithms 2 and 3 with an increasing batch size (2). Then, the following holds:*

$$\frac{1}{T}\sum_{t=0}^{T-1}\mathbb{E}\left[\mathcal{G}(\boldsymbol{\theta}_t)\right] \leq \frac{f(\boldsymbol{\theta}_0) - f(\boldsymbol{\theta}^\star)}{\gamma T} + \frac{D\sigma}{T}\frac{E\sqrt{\lambda}}{\sqrt{b}(\sqrt{\lambda}-1)} + \frac{LD^2\gamma}{2} = \mathcal{O}\left(\frac{1}{T} + \gamma\right),$$

*where $\boldsymbol{\theta}^\star := \underset{\boldsymbol{\theta}\in\Omega}{\operatorname{argmin}} \ f(\boldsymbol{\theta})$.*

## 3.2 Convergence analyses of SFW algorithms using a decaying learning rate

The following are convergence analyses of Algorithm 3 with a decaying learning rates (I), (II), and (III). The proofs of Theorems 3.3 and 3.4 are given in Appendices C and D.

**Theorem 3.3** (**Decaying Learning Rate (I, II, III) and Constant Batch Size**). *Suppose that Assumption 2.1 and the monotone decreasing property of $(\gamma_t)_{t\in\mathbb{N}}$ hold and consider the sequence $(\boldsymbol{\theta}_t)_{t\in\mathbb{N}}$ generated by each of Algorithms 2 and 3 with a constant batch size $b_t := b$. Then, the following is true:*

$$\frac{1}{T}\sum_{t=0}^{T-1}\mathbb{E}\left[\mathcal{G}(\boldsymbol{\theta}_t)\right] \leq \frac{2\max\{\bar{f}, |f(\boldsymbol{\theta}^\star)|\}}{T\gamma_{T-1}} + \frac{D\sigma}{\sqrt{b}} + \frac{LD^2}{2T}\sum_{t=0}^{T-1}\gamma_t,$$

*where $\boldsymbol{\theta}^\star := \underset{\boldsymbol{\theta}\in\Omega}{\operatorname{argmin}} \ f(\boldsymbol{\theta})$, and $\bar{f}$ is an upper bound of $f$.*

(i) *If we use $\gamma_t = \frac{1}{(t+1)}$, then*

$$\frac{1}{T}\sum_{t=0}^{T-1}\mathbb{E}\left[\mathcal{G}(\boldsymbol{\theta}_t)\right] \leq 2\max\{\bar{f}, |f(\boldsymbol{\theta}^\star)|\} + \frac{D\sigma}{\sqrt{b}} + \frac{LD^2(1+\log T)}{2T} = \mathcal{O}\left(\frac{\log T}{T} + \frac{1}{\sqrt{b}} + C\right).$$

(ii) *If we use $\gamma_t = \frac{1}{(t+1)^a}(a \in [\frac{1}{2}, 1))$, then*

$$\frac{1}{T}\sum_{t=0}^{T-1}\mathbb{E}\left[\mathcal{G}(\boldsymbol{\theta}_t)\right] \leq \frac{2\max\{\bar{f}, |f(\boldsymbol{\theta}^\star)|\}}{T^{1-a}} + \frac{D\sigma}{\sqrt{b}} + \frac{LD^2}{2(1-a)T^a} = \mathcal{O}\left(\frac{1}{T^{\min\{1-a,a\}}} + \frac{1}{\sqrt{b}}\right).$$

(iii) *If we use $\gamma_t = (\underbrace{\gamma, \gamma, \cdots, \gamma}_{K}, \underbrace{\eta\gamma, \eta\gamma, \cdots, \eta\gamma}_{K}, \cdots, \underbrace{\eta^{P-1}\gamma, \eta^{P-1}\gamma, \cdots, \eta^{P-1}\gamma}_{K})$, then*

$$\frac{1}{T}\sum_{t=0}^{T-1}\mathbb{E}\left[\mathcal{G}(\boldsymbol{\theta}_t)\right] \leq \frac{2\max\{\bar{f}, |f(\boldsymbol{\theta}^\star)|\}}{T\underline{\gamma}} + \frac{D\sigma}{\sqrt{b}} + \frac{LD^2}{2T}\frac{K\gamma}{1-\eta} = \mathcal{O}\left(\frac{1}{T} + \frac{1}{\sqrt{b}}\right),$$

*where $\gamma > 0, \eta \in (0, 1), C > 0$ is a constant, $\underline{\gamma}$ is the lower bound of $\gamma_t$ defined as (1), and for a fixed natural number $K$, $PK = T$.*

**Theorem 3.4** (**Decaying Learning Rate (I, II, III) and Increasing Batch Size**). *Suppose that Assumption 2.1 and the monotone decreasing property of $(\gamma_t)_{t\in\mathbb{N}}$ hold and consider the sequence $(\boldsymbol{\theta}_t)_{t\in\mathbb{N}}$ generated by each of Algorithms 2 and 3 with an increasing batch size (2).Then, the following is true:*

$$\frac{1}{T}\sum_{t=0}^{T-1}\mathbb{E}\left[\mathcal{G}(\boldsymbol{\theta}_t)\right] \leq \frac{2\max\{\bar{f}, |f(\boldsymbol{\theta}^\star)|\}}{T\gamma_{T-1}} + \frac{D\sigma}{T}\frac{E\sqrt{\lambda}}{\sqrt{b}(\sqrt{\lambda}-1)} + \frac{LD^2}{2T}\sum_{t=0}^{T-1}\gamma_t,$$

*where $\boldsymbol{\theta}^\star := \underset{\boldsymbol{\theta}\in\Omega}{\arg\min}\, f(\boldsymbol{\theta})$, and $\bar{f}$ is an upper bound of $f$.*

(i) *If we use $\gamma_t = \frac{1}{(t+1)}$, then*

$$\frac{1}{T}\sum_{t=0}^{T-1}\mathbb{E}\left[\mathcal{G}(\boldsymbol{\theta}_t)\right] \leq 2\max\{\bar{f}, |f(\boldsymbol{\theta}^\star)|\} + \frac{D\sigma}{T}\frac{E\sqrt{\lambda}}{\sqrt{b}(\sqrt{\lambda}-1)} + \frac{LD^2(1+\log T)}{T} = \mathcal{O}\left(\frac{\log T}{T} + C\right).$$

(ii) *If we use $\gamma_t = \frac{1}{(t+1)^a} (a \in [\frac{1}{2}, 1))$, then*

$$\frac{1}{T}\sum_{t=0}^{T-1}\mathbb{E}\left[\mathcal{G}(\boldsymbol{\theta}_t)\right] \leq \frac{2\max\{\bar{f}, |f(\boldsymbol{\theta}^\star)|\}}{T^{1-a}} + \frac{D\sigma}{T}\frac{E\sqrt{\lambda}}{\sqrt{b}(\sqrt{\lambda}-1)} + \frac{LD^2}{2(1-a)T^a} = \mathcal{O}\left(\frac{1}{T^{\min\{1-a,a\}}}\right).$$

(iii) *If we use $\gamma_t = (\underbrace{\gamma, \gamma, \cdots, \gamma}_{K}, \underbrace{\eta\gamma, \eta\gamma, \cdots, \eta\gamma}_{K}, \cdots, \underbrace{\eta^{P-1}\gamma, \eta^{P-1}\gamma, \cdots, \eta^{P-1}\gamma}_{K}), then*

$$\frac{1}{T}\sum_{t=0}^{T-1}\mathbb{E}\left[\mathcal{G}(\boldsymbol{\theta}_t)\right] \leq \frac{2\max\{\bar{f}, |f(\boldsymbol{\theta}^\star)|\}}{T\underline{\gamma}} + \frac{D\sigma}{T}\frac{E\sqrt{\lambda}}{\sqrt{b}(\sqrt{\lambda}-1)} + \frac{LD^2}{2T}\frac{K\gamma}{1-\eta} = \mathcal{O}\left(\frac{1}{T}\right).$$

*where $\gamma > 0, \eta \in (0, 1), C > 0$ is a constant, $\underline{\gamma}$ is the lower bound of $\gamma_t$ defined as (1), and for a fixed natural number $K$, $PK = T$.*

Our theorems were analyzed for the sequence $(\boldsymbol{\theta}_t)$ generated by SFWM (Algorithm 3), but the momentum factor $\beta$ does not appear in the upper bound for the Frank Wolfe gap, thanks to the key lemma (Lemma A.3). Therefore, Theorems 3.1-3.4 also hold for the sequence $(\boldsymbol{\theta}_t)$ generated by SFW (Algorithm 2).

### 3.3 Numerical Results

To verify the performance of Algorithms 2 and 3, we conducted numerical experiments on training ResNet18 (He et al., 2016) with the CIFAR100 (Krizhevsky, 2009) dataset under the $L_2$ constraint. The training took 200 epochs. The initial learning rate $\gamma_0$ was 0.1 for all algorithms and the momentum factor $\beta$ was 0.9 for SFWM. The radius of the $L_2$ constraint was set to 300. Figure 1 plots the top-1 test accuracy and loss function values with the SFW methods. SFW (constant bs), SFWM (constant bs), SVFW (Reddi et al., 2016), SAGAFW (Reddi et al., 2016), and Mokhtari's SFW (Mokhtari et al., 2020) used a constant batch size of $2^{10}$ and a constant learning rate $\gamma_t = \gamma_0 = 0.1$, i.e., this setting is based on Theorem 3.1. SFW (increasing bs) and SFWM (increasing bs) used batch sizes that quadruple every 40 epochs; i.e., this setting is based on Theorem 3.2. We tuned Adam with learning rates {0.0001, 0.001, 0.01}, $\beta_1$ values {0, 0.5, 0.7, 0.9} and $\beta_2$ values {0.9, 0.99, 0.999}. Based on our tuning results, we selected a learning rate of 0.001, and betas of $(0.9, 0.999)$ for our experiments. Adam (Kingma & Ba, 2015) used 0.001 of learning rate and $(0.9, 0.999)$ of betas. Note that Mokhtari's SFW have momentum factor that varies with iteration $t$. This is different from SFWM (Algorithm 3) where the momentum factor is fixed. The rising curve is test accuracy and the falling curve is the training loss function.

Figure 2 similarly plots test accuracy and the loss function values for a batch size of $2^{10}$ and a decaying learning rate that halves every 40 epochs; i.e., this setting is based on Theorem 3.3(ii). SFW (increasing bs)

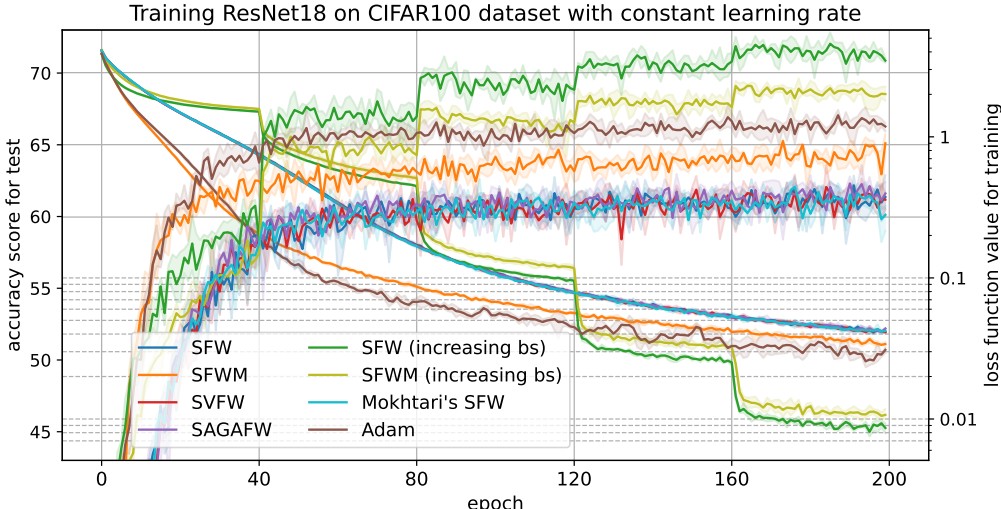

Figure 1: Accuracy score for the testing and loss function value for training versus the number of epochs in training ResNet18 on the CIFAR100 dataset with the $L_2$ constraint and a constant learning rate. The solid line represents the mean value, and the shaded area represents the maximum and minimum over three runs. The batch size was increased every 40 epochs as $[8, 32, 128, 512, 2048]$ for SFW (increasing bs) and SFWM (increasing bs). (bs: batch size).

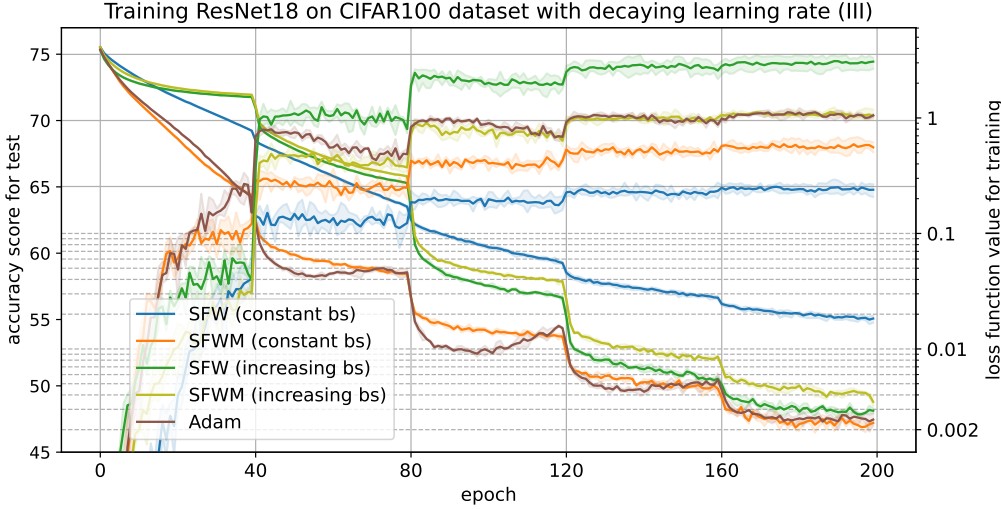

Figure 2: Accuracy score for the testing and loss function value for training versus the number of epochs in training ResNet18 on the CIFAR100 dataset with the $L_2$ constraint and a decaying learning rate. The solid line represents the mean value, and the shaded area represents the maximum and minimum over three runs. For SFW and SFWM, the learning rate was decreased every 40 epochs as $[0.1, 0.05, 0.025, 0.0125, 0.00625]$ and batch size was fixed at $2^{10}$. The batch size was increased every 40 epochs as $[8, 32, 128, 512, 2048]$ and the learning rate was decreased using the same rule for SFW (increasing bs) and SFWM (increasing bs). (bs: batch size).

and SFWM (increasing bs) used batch sizes that quadruple every 40 epochs; i.e., this setting is based on Theorem 3.4(ii).

Both figures show that SFWM (constant bs) achieve higher test accuracy and lower loss function values than those of SFW (constant bs). Note, however, that this is due to the large batch size. We have performed similar experiments with different batch sizes and observed that the exact opposite results are obtained when the batch size is small (see Figures 3 and 4 in Appendix E.1). Thus, for SFW, adding momentum helps when

the batch size is large, but has the opposite effect when the batch size is small. Note that we only consider a fixed momentum factor. This is the same phenomenon observed in SGD and SGD with momentum that some previous studies have observed experimentally (Shallue et al., 2019; Jelassi & Li, 2022; Kunstner et al., 2023). In addition, Figures 1 and 2 show that, for all methods, the decaying learning rate achieves higher test accuracy and lower loss function values than the constant learning rate does. This finding is theoretically supported by Theorems 3.1 and 3.3(ii) (see also Table 2). Furthermore, Figures 1 and 2 show that, for SFW, the increasing batch size achieves higher test accuracy and lower loss function values than the constant batch size does. For SFWM, however, using a decaying learning rate and increasing batch size improves the test accuracy, but worsens the loss function value. This cannot be explained by our theorem and is a limitation of our theory.

We also performed similar experiments for decaying learning rates (I) $\gamma_t := \frac{1}{t+1}$ based on Theorem 3.3(i) and (II) $\gamma_t := \frac{1}{\sqrt{t+1}}$ based on Theorem 3.3(ii) (see Figures 5 and 6 in Appendix E.1) and observed their limitations.

In fact, the ResNet18 training we have performed in this section does not need to be constrained, and unconstrained optimization algorithms such as SGD are sufficient.

## 4 Application to Adversarial Attacks

In this section, we explain that the optimization problem for the adversarial attack is an instance of a constrained non-convex optimization problem and experimentally verify whether this attack succeeds with FW methods.

Deep neural networks are vulnerable to adversarial examples (Szegedy et al., 2014; Goodfellow et al., 2015), which are generated by adding perturbations to real images that are too small for the human eye to perceive. Improving the robustness of classifiers to adversarial examples has become one of the most studied topics in the machine learning community. Adversarial training (Goodfellow et al., 2015), which includes adversarial examples in the training data, is effective in improving robustness. Generating adversarial examples is also important for robust model development because adversarial examples can be used to evaluate the robustness of a model. An adversarial attack is a method to create adversarial examples, and various algorithms have been proposed. Depending on the amount of information an attacker has access to, adversarial attacks can be divided into white-box attacks (Szegedy et al., 2014; Goodfellow et al., 2015) and black-box attacks (Papernot et al., 2016; Chen et al., 2017). In a white-box attack, the attacker has access to the complete information including the weights of the target model, while in a black-box attack, the adversary has access only to the inputs and outputs of the target model. In this paper, we focus on white-box attacks.

### 4.1 White-box attack

For white-box attacks, Szegedy et al. (2014) proposed to use the box-constrained limited-memory Broyden-Fletcher-Goldferb-Shanno (L-BFGS) algorithm. Goodfellow et al. (2015) proposed the fast gradient sign method (FGSM) to overcome the speed limitation of L-BFGS. Kurakin et al. (2016) proposed an iterative-FGSM (I-FGSM) algorithm that iterates over the one-iteration FGSM algorithm. Madry et al. (2018) showed that I-FGSM with the $L_\infty$ norm is approximately equivalent to projected gradient descent (PGD). PGD is the most popular attack because it is computationally inexpensive and has been successful in many cases. Auto-PGD (APGD) (Croce & Hein, 2020) is an attack that searches for perturbations while dynamically varying the learning rate of PGD. In this paper, FW is used to attack. Note that we are not the first to use FW-type algorithms in adversarial attacks. Several previous studies tackled adversarial attacks using the Frank Wolfe algorithm or its variants. Chen et al. (2020a) applied Frank Wolfe with momentum factor to white-box and black-box attacks against a non-robust model and showed that FW has better attack performance than PGD. Several previous results studies the effectiveness of FW-type zeroth order optimization algorithms against black-box attack (Chen et al., 2020a; Sahu & Kar, 2020; Huang et al., 2020a). Imtiaz et al. (2022) proposed the Sparse Adversarial and Interpretable Attack Framework (SAIF), an attack method that minimizes magnitude and sparsity of perturbations using FW, and showed that it outperforms conventional methods, especially when the perturbation constraints are tight. Kazemi et al. (2021) consider

several structured constraints different from the traditional $L_p$ norm constraints, and standard FW is used to generate the adversarial example.

## 4.2 Problem setting

Given a training dataset $\mathcal{S} = \{(\boldsymbol{x}_i, \boldsymbol{y}_i) | \boldsymbol{x}_i \in \mathbb{R}^d, \boldsymbol{y}_i \in \mathbb{R}^c\}_{i=0}^{n-1}$ drawn i.i.d. from the distribution, where $c \in \mathbb{N}$ is the number of classes in the training dataset, consider a classification with a neural network $g(\boldsymbol{x}, \boldsymbol{\theta})$. Here, $\boldsymbol{x}_i \in \mathbb{R}^d$ is the real image, $\boldsymbol{y}_i$ is the label for $\boldsymbol{x}_i$, and $\boldsymbol{\theta}$ is the parameter of the model $g$. Let $g(\boldsymbol{x})$ be the predicted labels of the classifier and $f_i(g(\boldsymbol{x}_i), \boldsymbol{y}_i)$ be the loss function. The parameters of the neural network are updated to minimize $f(\boldsymbol{\theta}) := \frac{1}{n} \sum_{i=0}^{n-1} f_i(g(\boldsymbol{x}_i), \boldsymbol{y}_i)$. When training is completed, the classifier $g$ will be able to classify the input image $\boldsymbol{x}$ with high accuracy. We aim to skew the predictions of the trained classifier by adding a miniscule amount of noise to the input image $\boldsymbol{x}$. Given a distance function $d(\cdot, \cdot)$ and a positive number $\epsilon > 0$, the generated adversarial example $\boldsymbol{x}_{\mathrm{adv}}$ can be expressed as $\boldsymbol{x}_{\mathrm{adv}} \in \{\hat{\boldsymbol{x}}_i \mid g(\hat{\boldsymbol{x}}_i) \neq y_i, d(\hat{\boldsymbol{x}}_i, \boldsymbol{x}_i) \leq \epsilon\}$. The adversarial attack can be formulated as follows:

$$\max_{\hat{\boldsymbol{x}} \in \mathbb{R}^d} f(g(\hat{\boldsymbol{x}}), \boldsymbol{y}) \text{ s.t. } d(\hat{\boldsymbol{x}}, \boldsymbol{x}) \leq \epsilon.$$

This is an example of Problem 2.1, and the constraint set $\Omega$ in Problem 2.1 can be expressed as $\Omega = \{\hat{\boldsymbol{x}} \in \mathbb{R}^d \mid d(\hat{\boldsymbol{x}}, \boldsymbol{x}) \leq \epsilon\}$, which satisfies Assumption (A1). In experiments, the Euclidean norm or maximum norm is often chosen as the distance function $d(\cdot, \cdot)$. $\epsilon$ is important because it controls the amount of noise that can be added to the real images. The larger $\epsilon$ is, the more it can distort the real image and increase the success rate of the attack. However, since we want to add noise that is imperceptible to humans, we should use a small value of $\epsilon$. In experiments, $\epsilon$ is often set to a small value such as $8/255$ or $4/255$. See Section 4.5 for the detailed experimental setup.

## 4.3 Auto-Frank Wolfe Attack

We propose the Auto-Frank Wolfe (AFW) attack as a new adversarial attack derived from the FW approach. The proposed scheme is summarized in Algorithm 4.

---

**Algorithm 4** Auto-Frank Wolfe

---

**Require:** $f, \Omega, \boldsymbol{x}_0, \gamma_0, N_{\mathrm{iter}}, W = \{w_0, \cdots, w_n\}$

  $\boldsymbol{x}_{\mathrm{adv}} \leftarrow \boldsymbol{x}_0, \boldsymbol{m}_{-1} \leftarrow \boldsymbol{0}$

  **for** $t = 0$ to $N_{\mathrm{iter}} - 1$ **do**

    $\boldsymbol{v}_t = \underset{\boldsymbol{v} \in \Omega}{\mathrm{argmax}} \langle \boldsymbol{v}, \nabla f(\boldsymbol{x}_t) \rangle$

    $\boldsymbol{d}_t = \boldsymbol{v}_t - \boldsymbol{x}_t$

    $\boldsymbol{x}_{t+1} = \boldsymbol{x}_t + \gamma_t \boldsymbol{d}_t$

    **if** $f(\boldsymbol{x}_{t+1}) > f(\boldsymbol{x}_{\mathrm{adv}})$ **then**

      $\boldsymbol{x}_{\mathrm{adv}} \leftarrow \boldsymbol{x}_{t+1}$

    **end if**

    **if** $t \in W$ **then**

      **if** Condition (i) or (ii) is satisfied **then**

        $\gamma_{t+1} \leftarrow \gamma_t / 2$

        $\boldsymbol{x}_{t+1} \leftarrow \boldsymbol{x}_{\mathrm{adv}}$

      **end if**

    **end if**

  **end for**

---

In the part of Algorithm 4 that calculates $\boldsymbol{v}_t$, $\langle \boldsymbol{v}, \nabla f(\boldsymbol{x}_t) \rangle$ is used instead of $\langle \boldsymbol{v}, -\nabla f(\boldsymbol{x}_t) \rangle$. This is because we are trying to increase the function value. In Algorithm 4, the method proposed in APGD is used for step size selection (Croce & Hein, 2020). The initial step size $\gamma_0$ is set to $2\epsilon$. When the number of iterations reaches the pre-calculated checkpoint $w_j \in W$, the step size $\gamma_t$ is halved if either of the following two conditions is satisfied. These conditions indicate that the attack is not going well.

*(i)* $N_c < \rho \cdot (w_j - w_{j-1})$,

*(ii)* $\gamma_{w_{j-1}} = \gamma_{w_j}$ and $f_{\max}^{(w_{j-1})} = f_{\max}^{(w_j)}$,

where $N_c := \#\{i = w_{j-1}, \cdots, w_j - 1 \mid f(\boldsymbol{x}_{t+1}) > f(\boldsymbol{x}_t)\}$, $\rho > 0$, and $f_{\max}^{(t)} := \max\{f(\boldsymbol{x}_k) \mid k = 0, \cdots, t\}$.

Regarding the convergence guarantee of AFW, we see from Theorem 3.1 that AFW has a convergence rate of $\mathcal{O}\left(\frac{1}{N_{\text{iter}}} + \gamma_0\right)$ if the attacks by AFW are consistently successful, since AFW uses the same learning rate for all iterations. Note, however, that since AFW is a deterministic algorithm, the term involving batch size b disappears in Theorem 3.1. On the other hand, if the AFW attack is not effective, the point sequence is pulled back to the checkpoints and the learning rate is halved. For every $n$ checkpoints, the number of cases to consider is $2^n - 1$. This makes it difficult to theoretically show the convergence of AFW, and this is also true for APGD and ACG.

### 4.4 Adversarial robustness

Many methods have been proposed to improve the robustness of models, such as gradient regularization (Ross & Doshi-Velez, 2018), curvature regularization (Moosavi-Dezfooli et al., 2018; Huang et al., 2020c), randomized smoothing (Cohen et al., 2019), local linearization (Qin et al., 2019), and adversarial training (Goodfellow et al., 2015; Madry et al., 2018; Zhang et al., 2019b; Wu et al., 2020; Wang et al., 2020; Zhang et al., 2020; Huang et al., 2021b; Balaji et al., 2019; Zhang et al., 2021; Pang et al., 2022; Wang et al., 2023). Among these methods, adversarial training is the most standard one to improve adversarial robustness. It improves robustness by learning adversarial examples in addition to the usual training data. For non-robust models, almost all methods can achieve a classification accuracy of 0% (for example, see (Chen et al., 2020a, Table 1 and 2)). So in this paper, we evaluate the attack performance of FW algorithms by attacking robust models trained with adversarial training and compare it with PGD and APGD.

### 4.5 Numerical Results

To evaluate the attack performance of AFW, we conducted an experiment comparing the performance of APGD and AFW against state-of-the-art robust models listed in RobustBench (Croce et al., 2021). Table 3 shows the results of APGD and AFW intercepting the classification task of the CIFAR100 dataset with several robust models. The loss function was the cross entropy loss. The constraint set was $L_\infty$ with diameter $\epsilon = 8/255$ and the attack was executed over the course of 100 steps. In all experiments, both APGD and AFW use an initial learning rate of $4\epsilon$. This is the default value in the APGD implementation (Croce & Hein, 2020). Clean accuracy refers to the classification accuracy achieved by the model before the attack, while adversarial accuracy refers to the classification accuracy achieved by the model after the attack. Therefore, a lower adversarial accuracy implies a more successful attack. Table 3 shows that AFW has an attack performance almost equal to that of APGD, a state-of-the-art attack method. We also performed PGD, FW, APGD, and AFW attacks on the image classification task of the CIFAR10, CIFAR100, and ImageNet dataset and obtained similar results (see Tables 5-8 in Appendix E.2).

In addition, Table 4 shows the results of APGD and AFW intercepting the classification task of the CIFAR10, CIFAR100, and ImageNet dataset with robust model proposed by (Jiang et al., 2023). The adversarial perturbations is bounded by an $L_1$ norm and the threshold $\epsilon$ for each dataset follows their setting. Note that Fast-EG-$L_1$ is the method proposed by (Jiang et al., 2023) and nuclear norm adversarial training (NuAT) is the method proposed by (Sriramanan et al., 2021). Table 4 also shows that AFW has an attack performance almost equal to that of APGD.

Throughout all of the experimental results, the difference in performance between AFW and APGD is less than 1 percent. This is also the case in previous study that proposed ACG (Yamamura et al., 2022) and may be due to the very small size of the constraint set in an adversarial attack. In terms of computational complexity and time, AFW performs similarly to APGD, and we found no advantage of AFW over APGD. Although we have indeed shown that AFW is effective for adversarial attacks, the choice of optimization method may not be so important for adversarial attacks.

Table 3: Adversarial accuracy achieved by APGD and AFW. CIFAR100 dataset/$L_\infty$ with $\epsilon = 8/255$

| paper | Architecture | clean accuracy | APGD | AFW |
|---|---|---|---|---|
| (Wang et al., 2023) | WideResNet-70-16 | 75.22 | 48.11 | 48.16 |
| (Wang et al., 2023) | WideResNet-28-10 | 72.58 | 44.05 | 44.12 |
| (Debenedetti et al., 2023) | XCiT-L12 | 70.76 | 38.97 | 39.02 |
| (Rebuffi et al., 2021) | WideResNet-70-16 | 63.56 | 38.28 | 38.31 |
| (Debenedetti et al., 2023) | XCiT-M12 | 69.21 | 38.69 | 38.77 |
| (Pang et al., 2022) | WideResNet-70-16 | 65.56 | 36.66 | 36.71 |
| (Debenedetti et al., 2023) | XCiT-S12 | 67.34 | 37.08 | 37.06 |
| (Rebuffi et al., 2021) | WideResNet-28-16 | 62.41 | 35.73 | 35.8 |
| (Jia et al., 2022) | WideResNet-34-20 | 67.31 | 36.46 | 36.62 |
| (Addepalli et al., 2022a) | WideResNet-34-10 | 68.75 | 36.84 | 36.92 |
| (Cui et al., 2021) | WideResNet-34-10 | 62.97 | 37.05 | 37.17 |
| (Sehwag et al., 2022) | WideResNet-34-10 | 65.93 | 35.77 | 35.89 |
| (Pang et al., 2022) | WideResNet-28-10 | 63.66 | 35.25 | 35.26 |
| (Jia et al., 2022) | WideResNet-34-10 | 64.89 | 35.28 | 35.53 |
| (Addepalli et al., 2022b) | WideResNet-34-10 | 65.73 | 35.64 | 36.07 |
| (Cui et al., 2021) | WideResNet-34-20 | 62.55 | 34.08 | 34.27 |
| (Cui et al., 2021) | WideResNet-34-10 | 60.64 | 33.99 | 34.10 |
| (Rade & Moosavi-Dezfooli, 2021) | PreActResNet-18 | 61.50 | 32.58 | 32.61 |
| (Wu et al., 2020) | WideResNet-34-10 | 60.38 | 33.25 | 33.34 |
| (Rebuffi et al., 2021) | PreActResNet-18 | 56.87 | 31.77 | 31.8 |
| (Hendrycks et al., 2019) | WideResNet-28-10 | 59.23 | 32.88 | 33.01 |
| (Addepalli et al., 2022a) | ResNet-18 | 65.45 | 33.55 | 33.68 |
| (Cui et al., 2021) | WideResNet-34-10 | 70.25 | 29.98 | 30.02 |
| (Addepalli et al., 2022b) | PreActResNet-18 | 62.02 | 32.88 | 32.94 |
| (Chen et al., 2022) | WideResNet-34-10 | 62.15 | 30.98 | 31.13 |
| (Rice et al., 2020) | PreActResNet-18 | 53.83 | 20.63 | 20.57 |

Table 4: Adversarial accuracy achieved by APGD and AFW with $L_1$ constraints.

| dataset (threshold) | Architecture (method) | clean accuracy | APGD | AFW |
|---|---|---|---|---|
| CIFAR10 ($\epsilon = 12$) | PreActResNet-18 (Fast-EG-$L_1$) | 76.22 | 51.87 | 52.03 |
| CIFAR10 ($\epsilon = 12$) | PreActResNet-18 (Fast-EG-$L_1$+NuAT) | 73.73 | 53.22 | 53.27 |
| CIFAR100 ($\epsilon = 6$) | PreActResNet-18 (Fast-EG-$L_1$) | 59.43 | 39.56 | 39.61 |
| CIFAR100 ($\epsilon = 6$) | PreActResNet-18 (Fast-EG-$L_1$+NuAT) | 58.50 | 41.73 | 41.65 |
| ImageNet100 ($\epsilon = 72$) | ResNet34 (Fast-EG-$L_1$) | 67.62 | 48.80 | 48.76 |
| ImageNet100 ($\epsilon = 72$) | ResNet34 (Fast-EG-$L_1$+NuAT) | 62.34 | 50.50 | 50.48 |

## 5 Conclusions

We provided a practical convergence analysis of SFW and SFW with momentum with a constant or decaying learning rate for solving constrained nonconvex optimization problems. In our analysis, the learning rate and the batch size are independent of unknown parameters and are experimentally realistic. We showed that when the momentum factor is zero or fixed, the Frank Wolfe gap has a convergence rate of $\mathcal{O}(1/T)$ only when we decrease the learning rate and increase the batch size. Our numerical experiments show that SFW with momentum outperforms SFW in both test accuracy and loss function value when the batch size is large and that using a decaying learning rate and increasing batch size achieves higher test accuracy and lower loss function values than does using a constant learning rate in image classification tasks with ResNet18. We also showed experimentally that FW algorithms perform as well as PGD in adversarial attacks and that our proposed AFW performs as well as APGD.

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

## A  Proposition and Lemmas

Propositions A.1 and Lemma A.2 are general results with no novelty.

**Proposition A.1.** *For all $\boldsymbol{x}, \boldsymbol{y} \in \mathbb{R}^d$ and all $\alpha \in \mathbb{R}$, the following holds:*

$$\|\alpha\boldsymbol{x} + (1-\alpha)\boldsymbol{y}\|^2 = \alpha\|\boldsymbol{x}\|^2 + (1-\alpha)\|\boldsymbol{y}\|^2 - \alpha(1-\alpha)\|\boldsymbol{x} - \boldsymbol{y}\|^2.$$

*Proof.* Since $2\langle \boldsymbol{x}, \boldsymbol{y}\rangle = \|\boldsymbol{x}\|^2 + \|\boldsymbol{y}\|^2 - \|\boldsymbol{x} - \boldsymbol{y}\|^2$ holds, for all $\boldsymbol{x}, \boldsymbol{y} \in \mathbb{R}^d$ and all $\alpha \in \mathbb{R}$,

$$
\begin{aligned}
\|\alpha\boldsymbol{x} + (1-\alpha)\boldsymbol{y}\|^2 &= \alpha\|\boldsymbol{x}\|^2 + 2\alpha(1-\alpha)\langle \boldsymbol{x}, \boldsymbol{y}\rangle + (1-\alpha)^2\|\boldsymbol{y}\|^2 \\
&= \alpha\|\boldsymbol{x}\|^2 + \alpha(1-\alpha)(\|\boldsymbol{x}\|^2 + \|\boldsymbol{y}\|^2 - \|\boldsymbol{x} - \boldsymbol{y}\|^2) + (1-\alpha)^2\|\boldsymbol{y}\|^2 \\
&= \alpha\|\boldsymbol{x}\|^2 + (1-\alpha)\|\boldsymbol{y}\|^2 - \alpha(1-\alpha)\|\boldsymbol{x} - \boldsymbol{y}\|^2.
\end{aligned}
$$

This completes the proof. $\qquad\square$

**Lemma A.1.** *Suppose that (A1)-(A4) hold and consider Algorithm 3. Then, the following holds:*

$$\gamma_t \mathbb{E}\left[\mathcal{G}(\boldsymbol{\theta}_t)\right] \leq \mathbb{E}\left[f(\boldsymbol{\theta}_t)\right] - \mathbb{E}\left[f(\boldsymbol{\theta}_{t+1})\right] + \frac{D\sigma\gamma_t}{\sqrt{b_t}} + \frac{LD^2\gamma_t^2}{2}.$$

*Proof.* Let $t \in \mathbb{N}$. (A2) and the definition of $\boldsymbol{\theta}_{t+1}$ guarantee that

$$
\begin{aligned}
f(\boldsymbol{\theta}_{t+1}) &\leq f(\boldsymbol{\theta}_t) + \langle \nabla f(\boldsymbol{\theta}_t), \boldsymbol{\theta}_{t+1} - \boldsymbol{\theta}_t\rangle + \frac{L}{2}\|\boldsymbol{\theta}_{t+1} - \boldsymbol{\theta}_t\|^2 \\
&= f(\boldsymbol{\theta}_t) + \langle \nabla f(\boldsymbol{\theta}_t), (\boldsymbol{\theta}_t + \gamma_t(\boldsymbol{v}_t - \boldsymbol{\theta}_t)) - \boldsymbol{\theta}_t\rangle + \frac{L}{2}\|\boldsymbol{\theta}_{t+1} - \boldsymbol{\theta}_t\|^2 \\
&= f(\boldsymbol{\theta}_t) + \gamma_t\langle \nabla f(\boldsymbol{\theta}_t), \boldsymbol{v}_t - \boldsymbol{\theta}_t\rangle + \frac{L\gamma_t^2}{2}\|\boldsymbol{v}_t - \boldsymbol{\theta}_t\|^2 \\
&\leq f(\boldsymbol{\theta}_t) + \gamma_t\langle \nabla f(\boldsymbol{\theta}_t), \boldsymbol{v}_t - \boldsymbol{\theta}_t\rangle + \frac{LD^2\gamma_t^2}{2}.
\end{aligned}
$$

Let $\hat{\boldsymbol{v}}_t := \underset{\boldsymbol{v}\in\Omega}{\operatorname{argmax}}\langle \boldsymbol{v}, -\nabla f(\boldsymbol{\theta}_t)\rangle$ for all $t \in \mathbb{N}$. From the definition of $\boldsymbol{v}_t$, $\forall \boldsymbol{v} \in \Omega : \langle \nabla f_{\mathcal{S}_t}(\boldsymbol{\theta}_t), \boldsymbol{v}_t\rangle \leq \langle \nabla f_{\mathcal{S}_t}(\boldsymbol{\theta}_t), \boldsymbol{v}\rangle$. Then,

$$
\begin{aligned}
f(\boldsymbol{\theta}_{t+1}) &\leq f(\boldsymbol{\theta}_t) + \gamma_t\langle \boldsymbol{m}_t, \boldsymbol{v}_t - \boldsymbol{\theta}_t\rangle + \gamma_t\langle \nabla f(\boldsymbol{\theta}_t) - \boldsymbol{m}_t, \boldsymbol{v}_t - \boldsymbol{\theta}_t\rangle + \frac{LD^2\gamma_t^2}{2} \\
&\leq f(\boldsymbol{\theta}_t) + \gamma_t\langle \boldsymbol{m}_t, \hat{\boldsymbol{v}}_t - \boldsymbol{\theta}_t\rangle + \gamma_t\langle \nabla f(\boldsymbol{\theta}_t) - \boldsymbol{m}_t, \boldsymbol{v}_t - \boldsymbol{\theta}_t\rangle + \frac{LD^2\gamma_t^2}{2} \\
&= f(\boldsymbol{\theta}_t) + \gamma_t\langle \nabla f(\boldsymbol{\theta}_t), \hat{\boldsymbol{v}}_t - \boldsymbol{\theta}_t\rangle + \gamma_t\langle \boldsymbol{m}_t - \nabla f(\boldsymbol{\theta}_t), \hat{\boldsymbol{v}}_t - \boldsymbol{\theta}_t\rangle + \gamma_t\langle \nabla f(\boldsymbol{\theta}_t) - \boldsymbol{m}_t, \boldsymbol{v}_t - \boldsymbol{\theta}_t\rangle + \frac{LD^2\gamma_t^2}{2} \\
&= f(\boldsymbol{\theta}_t) + \gamma_t\langle \nabla f(\boldsymbol{\theta}_t), \hat{\boldsymbol{v}}_t - \boldsymbol{\theta}_t\rangle + \gamma_t\langle \boldsymbol{m}_t - \nabla f(\boldsymbol{\theta}_t), \hat{\boldsymbol{v}}_t - \boldsymbol{v}_t\rangle + \frac{LD^2\gamma_t^2}{2} \\
&= f(\boldsymbol{\theta}_t) - \gamma_t\mathcal{G}(\boldsymbol{\theta}_t) + \gamma_t\langle \boldsymbol{m}_t - \nabla f(\boldsymbol{\theta}_t), \hat{\boldsymbol{v}}_t - \boldsymbol{v}_t\rangle + \frac{LD^2\gamma_t^2}{2} \\
&\leq f(\boldsymbol{\theta}_t) - \gamma_t\mathcal{G}(\boldsymbol{\theta}_t) + \gamma_t\|\boldsymbol{m}_t - \nabla f(\boldsymbol{\theta}_t)\|D + \frac{LD^2\gamma_t^2}{2}.
\end{aligned}
$$

The last inequality follows from the Cauchy-Schwarz inequality and (A1). Taking the expectation with respect to $\xi_t$ on both sides and using Lemma A.3, we have

$$\mathbb{E}\left[f(\boldsymbol{\theta}_{t+1})\right] \leq \mathbb{E}\left[f(\boldsymbol{\theta}_t)\right] - \gamma_t\mathbb{E}\left[\mathcal{G}(\boldsymbol{\theta}_t)\right] + D\gamma_t\mathbb{E}\left[\|\boldsymbol{m}_t - \nabla f(\boldsymbol{\theta}_t)\|\right] + \frac{LD^2\gamma_t^2}{2} \qquad (3)$$

$$\leq \mathbb{E}\left[f(\boldsymbol{\theta}_t)\right] - \gamma_t\mathbb{E}\left[\mathcal{G}(\boldsymbol{\theta}_t)\right] + D\gamma_t\sqrt{\frac{\sigma^2}{b_t}} + \frac{LD^2\gamma_t^2}{2}.$$

This completes the proof. $\qquad\square$

**Lemma A.2.** *Algorithm 3 has the property that, under (A3)(ii) and (A4), for all $t \in \mathbb{N}$,*

$$\mathbb{E}\left[\|\nabla f_{\mathcal{S}_t}(\boldsymbol{\theta}_t) - \nabla f(\boldsymbol{\theta}_t)\|^2\right] \leq \frac{\sigma^2}{b_t}.$$

*Proof.* (A3) and the definition of $\nabla f_{\mathcal{S}_t}(\boldsymbol{\theta}_t)$ guarantee that

$$
\begin{aligned}
\mathbb{E}\left[\|\nabla f_{\mathcal{S}_t}(\boldsymbol{\theta}_t) - \nabla f(\boldsymbol{\theta}_t)\|^2\right] &= \mathbb{E}\left[\left\|\frac{1}{b_t}\sum_{i=1}^{b_t}\mathsf{G}_{\xi_{t,i}}(\boldsymbol{\theta}_t) - \nabla f(\boldsymbol{\theta}_t)\right\|^2\right] \\
&= \mathbb{E}\left[\left\|\frac{1}{b_t}\sum_{i=1}^{b_t}\mathsf{G}_{\xi_{t,i}}(\boldsymbol{\theta}_t) - \frac{1}{b_t}\sum_{i=1}^{b_t}\nabla f(\boldsymbol{\theta}_t)\right\|^2\right] \\
&= \mathbb{E}\left[\left\|\frac{1}{b_t}\sum_{i=1}^{b_t}(\mathsf{G}_{\xi_{t,i}}(\boldsymbol{\theta}_t) - \nabla f(\boldsymbol{\theta}_t))\right\|^2\right] \\
&= \frac{1}{b_t^2}\mathbb{E}\left[\left\|\sum_{i=1}^{b_t}(\mathsf{G}_{\xi_{t,i}}(\boldsymbol{\theta}_t) - \nabla f(\boldsymbol{\theta}_t))\right\|^2\right] \\
&= \frac{1}{b_t^2}\mathbb{E}\left[\sum_{i=1}^{b_t}\|\mathsf{G}_{\xi_{t,i}}(\boldsymbol{\theta}_t) - \nabla f(\boldsymbol{\theta}_t)\|^2\right] \\
&\leq \frac{\sigma^2}{b_t}.
\end{aligned}
$$

This completes the proof. $\qquad\square$

**Lemma A.3.** *Algorithm 3 has the property that, under (A3)(ii) and (A4), for all $t \in \mathbb{N}$,*

$$\mathbb{E}\left[\|\boldsymbol{m}_t - \nabla f(\boldsymbol{\theta}_t)\|\right] \leq \sqrt{\frac{\sigma^2}{b_t}}.$$

*Proof.* The definition of $\boldsymbol{m}_t$ implies that

$$
\begin{aligned}
\|\boldsymbol{m}_t - \nabla f(\boldsymbol{\theta}_t)\|^2 &= \|\beta\boldsymbol{m}_{t-1} + (1-\beta)\nabla f_{\mathcal{S}_t}(\boldsymbol{\theta}_t) - \nabla f(\boldsymbol{\theta}_t)\|^2 \\
&= \|\beta(\boldsymbol{m}_{t-1} - \nabla f(\boldsymbol{\theta}_t)) + (1-\beta)(\nabla f_{\mathcal{S}_t} - \nabla f(\boldsymbol{\theta}_t))\|^2 \\
&= \beta^2\|\boldsymbol{m}_{t-1} - \nabla f(\boldsymbol{\theta}_t)\|^2 + (1-\beta)^2\|\nabla f_{\mathcal{S}_t}(\boldsymbol{\theta}_t) - \nabla f(\boldsymbol{\theta}_t)\|^2 \\
&\quad + 2\beta(1-\beta)\langle\nabla f_{\mathcal{S}_t}(\boldsymbol{\theta}_t) - \nabla f(\boldsymbol{\theta}_t), \boldsymbol{m}_{t-1} - \nabla f(\boldsymbol{\theta}_t)\rangle.
\end{aligned}
$$

Therefore, from Assumption (A3)(i) and $\beta < 1$, we obtain

$$\mathbb{E}\left[\|\boldsymbol{m}_t - \nabla f(\boldsymbol{\theta}_t)\|^2\right] = (1-\beta)^2\mathbb{E}\left[\|\nabla f_{\mathcal{S}_t}(\boldsymbol{\theta}_t) - \nabla f(\boldsymbol{\theta}_t)\|^2\right] + \beta^2\mathbb{E}\left[\|\boldsymbol{m}_{t-1} - \nabla f(\boldsymbol{\theta}_t)\|^2\right] \tag{4}$$

$$< (1-\beta)^2\mathbb{E}\left[\|\nabla f_{\mathcal{S}_t}(\boldsymbol{\theta}_t) - \nabla f(\boldsymbol{\theta}_t)\|^2\right] + \mathbb{E}\left[\|\boldsymbol{m}_{t-1} - \nabla f(\boldsymbol{\theta}_t)\|^2\right]. \tag{5}$$

On the other hand, Proposition A.1 guarantees that

$$\mathbb{E}\left[\|\boldsymbol{m}_t - \nabla f(\boldsymbol{\theta}_t)\|^2\right] = (1-\beta)\mathbb{E}\left[\|\nabla f_{\mathcal{S}_t}(\boldsymbol{\theta}_t) - \nabla f(\boldsymbol{\theta}_t)\|^2\right] + \beta\mathbb{E}\left[\|\boldsymbol{m}_{t-1} - \nabla f(\boldsymbol{\theta}_t)\|^2\right] \tag{6}$$

$$- \beta(1-\beta)\mathbb{E}\left[\|\boldsymbol{m}_{t-1} - \nabla f_{\mathcal{S}_t}(\boldsymbol{\theta}_t)\|^2\right]. \tag{7}$$

From (4) and (7), we have

$$\mathbb{E}\left[\|\boldsymbol{m}_{t-1} - \nabla f(\boldsymbol{\theta}_t)\|^2\right] = \mathbb{E}\left[\|\boldsymbol{m}_{t-1} - \nabla f_{\mathcal{S}_t}(\boldsymbol{\theta}_t)\|^2\right] - \mathbb{E}\left[\|\nabla f_{\mathcal{S}_t}(\boldsymbol{\theta}_t) - \nabla f(\boldsymbol{\theta}_t)\|^2\right] \tag{8}$$

$$\leq \mathbb{E}\left[\|\boldsymbol{m}_{t-1} - \nabla f_{\mathcal{S}_t}(\boldsymbol{\theta}_t)\|^2\right]. \tag{9}$$

Therefore, from (5) and (8), we obtain

$$\mathbb{E}\left[\|\boldsymbol{m}_t - \nabla f(\boldsymbol{\theta}_t)\|^2\right] \leq \beta(-2+\beta)\mathbb{E}\left[\|\nabla f_{\mathcal{S}_t}(\boldsymbol{\theta}_t) - \nabla f(\boldsymbol{\theta}_t)\|^2\right] + \mathbb{E}\left[\|\boldsymbol{m}_{t-1} - \nabla f_{\mathcal{S}_t}(\boldsymbol{\theta}_t)\|^2\right]. \tag{10}$$

Now, let us show that, for all $t \in \mathbb{N}$,

$$\mathbb{E}\left[\|\boldsymbol{m}_{t-1} - \nabla f_{\mathcal{S}_t}(\boldsymbol{\theta}_t)\|^2\right] \leq \beta(2-\beta)\mathbb{E}\left[\|\nabla f_{\mathcal{S}_t}(\boldsymbol{\theta}_t) - \nabla f(\boldsymbol{\theta}_t)\|^2\right]. \tag{11}$$

If (11) does not hold, there exists $t_0 \in \mathbb{N}$ such that

$$\mathbb{E}\left[\|\boldsymbol{m}_{t_0-1} - \nabla f_{\mathcal{S}_{t_0}}(\boldsymbol{\theta}_{t_0})\|^2\right] > \beta(2-\beta)\mathbb{E}\left[\|\nabla f_{\mathcal{S}_{t_0}}(\boldsymbol{\theta}_{t_0}) - \nabla f(\boldsymbol{\theta}_{t_0})\|^2\right],$$

which implies

$$\mathbb{E}\left[\|\nabla f_{\mathcal{S}_{t_0}}(\boldsymbol{\theta}_{t_0}) - \nabla f(\boldsymbol{\theta}_{t_0})\|^2\right] < \frac{1}{\beta(2-\beta)}\mathbb{E}\left[\|\boldsymbol{m}_{t_0-1} - \nabla f_{\mathcal{S}_{t_0}}(\boldsymbol{\theta}_{t_0})\|^2\right]. \tag{12}$$

Hence, from (10) and (12),

$$\mathbb{E}\left[\|\boldsymbol{m}_{t_0} - \nabla f(\boldsymbol{\theta}_{t_0})\|^2\right] < \beta(-2+\beta)\left\{\frac{1}{\beta(2-\beta)}\mathbb{E}\left[\|\boldsymbol{m}_{t_0-1} - \nabla f_{\mathcal{S}_{t_0}}(\boldsymbol{\theta}_{t_0})\|^2\right]\right\}$$
$$+ \mathbb{E}\left[\|\boldsymbol{m}_{t_0-1} - \nabla f_{\mathcal{S}_{t_0}}(\boldsymbol{\theta}_{t_0})\|^2\right]$$
$$= 0.$$

Since $\mathbb{E}\left[\|\boldsymbol{m}_{t_0} - \nabla f(\boldsymbol{\theta}_{t_0})\|^2\right] \geq 0$, there is a contradiction. Therefore, (11) holds for all $t \in \mathbb{N}$. Thus, Lemmas A.2, (4), (9), and (11) ensure that

$$\mathbb{E}\left[\|\boldsymbol{m}_t - \nabla f(\boldsymbol{\theta}_t)\|^2\right] \leq (1-\beta)^2\mathbb{E}\left[\|\nabla f_{\mathcal{S}_t}(\boldsymbol{\theta}_t) - \nabla f(\boldsymbol{\theta}_t)\|^2\right] + \beta^3(2-\beta)\mathbb{E}\left[\|\nabla f_{\mathcal{S}_t}(\boldsymbol{\theta}_t) - \nabla f(\boldsymbol{\theta}_t)\|^2\right]$$
$$= \left\{(1-\beta)^2 + \beta^3(2-\beta)\right\}\mathbb{E}\left[\|\nabla f_{\mathcal{S}_t}(\boldsymbol{\theta}_t) - \nabla f(\boldsymbol{\theta}_t)\|^2\right]$$
$$\leq \frac{\sigma^2}{b_t}.$$

This completes the proof. $\qquad\square$

**Lemma A.4.** *Let the batch size increase as* $b_t := (\underbrace{b, b, \cdots, b}_{E}, \underbrace{\lambda b, \lambda b, \cdots, \lambda b}_{E}, \cdots, \underbrace{\lambda^{Q-1}b, \lambda^{Q-1}b, \cdots, \lambda^{Q-1}b}_{E})$.

*Then, for all* $t \in \mathbb{N}$,

$$\sum_{t=0}^{T-1} \frac{1}{\sqrt{b_t}} \leq \frac{E\sqrt{\lambda}}{\sqrt{b}(\sqrt{\lambda}-1)},$$

*where* $\lambda > 1$ *and* $QE = T$.

*Proof.* From the definition of $b_t$,

$$\sum_{t=0}^{T-1} \frac{1}{\sqrt{b_t}} = \frac{E}{\sqrt{b}} + \frac{E}{\sqrt{\lambda b}} + \cdots + \frac{E}{\sqrt{\lambda^{Q-1}b}}$$
$$= \frac{E}{\sqrt{b}}\left(1 + \frac{1}{\sqrt{\lambda}} + \cdots + \frac{1}{\sqrt{\lambda^{Q-1}}}\right)$$
$$\leq \frac{E}{\sqrt{b}} \cdot \frac{\sqrt{\lambda}}{\sqrt{\lambda}-1}.$$

This completes the proof. $\qquad\square$

**Lemma A.5.** *Let the learning rate decrease as* $\gamma_t := (\underbrace{\gamma, \gamma, \cdots, \gamma}_{K}, \underbrace{\eta\gamma, \eta\gamma, \cdots, \eta\gamma}_{K}, \cdots, \underbrace{\eta^{P-1}\gamma, \eta^{P-1}\gamma, \cdots, \eta^{P-1}\gamma}_{K}).$

*Then, for all $t \in \mathbb{N}$,*

$$\sum_{t=0}^{T-1} \gamma_t \leq \frac{K\gamma}{1-\eta},$$

*where $\eta \in (0,1)$ and $PK = T$.*

*Proof.* From the definition of $\gamma_t$,

$$\sum_{t=0}^{T-1} \gamma_t = K\gamma + K\eta\gamma + \cdots + K\eta^{P-1}\gamma$$
$$= K\gamma \left(1 + \eta + \cdots + \eta^{P-1}\right)$$
$$\leq \frac{K\gamma}{1-\eta}.$$

This completes the proof. □

## B  Proofs of Theorems 3.1 and 3.2

The following is a convergence analysis of Algorithm 3 using a constant learning rate.

*Proof.* From Lemma A.1, we have

$$\gamma_t \mathbb{E}\left[\mathcal{G}(\boldsymbol{\theta}_t)\right] \leq \mathbb{E}\left[f(\boldsymbol{\theta}_t)\right] - \mathbb{E}\left[f(\boldsymbol{\theta}_{t+1})\right] + \frac{D\sigma\gamma_t}{\sqrt{b_t}} + \frac{LD^2\gamma_t^2}{2}.$$

Summing over $t$, we get

$$\sum_{t=0}^{T-1} \gamma_t \mathbb{E}\left[\mathcal{G}(\boldsymbol{\theta}_t)\right] \leq \mathbb{E}\left[f(\boldsymbol{\theta}_0)\right] - \mathbb{E}\left[f(\boldsymbol{\theta}_T)\right] + D\sigma \sum_{t=0}^{T-1} \frac{\gamma_t}{\sqrt{b_t}} + \frac{LD^2}{2} \sum_{t=0}^{T-1} \gamma_t^2$$
$$\leq f(\boldsymbol{\theta}_0) - f(\boldsymbol{\theta}^\star) + D\sigma \sum_{t=0}^{T-1} \frac{\gamma_t}{\sqrt{b_t}} + \frac{LD^2}{2} \sum_{t=0}^{T-1} \gamma_t^2.$$

Assumption (C1) guarantees that

$$\gamma \sum_{t=0}^{T-1} \mathbb{E}\left[\mathcal{G}(\boldsymbol{\theta}_t)\right] \leq f(\boldsymbol{\theta}_0) - f(\boldsymbol{\theta}^\star) + D\sigma\gamma \sum_{t=0}^{T-1} \frac{1}{\sqrt{b_t}} + \frac{LD^2}{2}T\gamma^2.$$

Hence,

$$\frac{1}{T} \sum_{t=0}^{T-1} \mathbb{E}\left[\mathcal{G}(\boldsymbol{\theta}_t)\right] \leq \frac{f(\boldsymbol{\theta}_0) - f(\boldsymbol{\theta}^\star)}{\gamma T} + \frac{D\sigma}{T} \sum_{t=0}^{T-1} \frac{1}{\sqrt{b_t}} + \frac{LD^2\gamma}{2}.$$

Then, for a constant batch size $b_t := b$, we have

$$\frac{1}{T} \sum_{t=0}^{T-1} \mathbb{E}\left[\mathcal{G}(\boldsymbol{\theta}_t)\right] \leq \frac{f(\boldsymbol{\theta}_0) - f(\boldsymbol{\theta}^\star)}{\gamma T} + \frac{D\sigma}{\sqrt{b}} + \frac{LD^2\gamma}{2}$$
$$= \mathcal{O}\left(\frac{1}{T} + \frac{1}{\sqrt{b}} + \gamma\right)$$

This completes the proof for Theorem 3.1.

Next, for an increasing batch size $b_t := (\underbrace{b, b, \cdots, b}_{E}, \underbrace{\lambda b, \lambda b, \cdots, \lambda b}_{E}, \cdots, \underbrace{\lambda^{Q-1} b, \lambda^{Q-1} b, \cdots, \lambda^{Q-1} b}_{E})$, from Lemma A.4, we have

$$\frac{1}{T} \sum_{t=0}^{T-1} \mathbb{E}\left[\mathcal{G}(\boldsymbol{\theta}_t)\right] \leq \frac{f(\boldsymbol{\theta}_0) - f(\boldsymbol{\theta}^\star)}{\gamma T} + \frac{D\sigma}{T} \frac{E\sqrt{\lambda}}{\sqrt{b}(\sqrt{\lambda} - 1)} + \frac{LD^2 \gamma}{2}$$

$$= \mathcal{O}\left(\frac{1}{T} + \gamma\right).$$

This completes the proof for Theorem 3.2. $\qquad\qquad\square$

## C Proof of Theorem 3.3

*Proof.* From Lemma A.1 and $b_t := b$, we have

$$\mathbb{E}\left[\mathcal{G}(\boldsymbol{\theta}_t)\right] \leq \frac{1}{\gamma_t} \left(\mathbb{E}\left[f(\boldsymbol{\theta}_t)\right] - \mathbb{E}\left[f(\boldsymbol{\theta}_{t+1})\right]\right) + \frac{D\sigma}{\sqrt{b}} + \frac{LD^2 \gamma_t}{2}.$$

Summing over $t$, we get

$$\sum_{t=0}^{T-1} \mathbb{E}\left[\mathcal{G}(\boldsymbol{\theta}_t)\right] \leq \underbrace{\sum_{t=0}^{T-1} \frac{1}{\gamma_t} \left(\mathbb{E}\left[f(\boldsymbol{\theta}_t)\right] - \mathbb{E}\left[f(\boldsymbol{\theta}_{t+1})\right]\right)}_{\Gamma_T} + \frac{D\sigma}{\sqrt{b}} T + \frac{LD^2}{2} \sum_{t=0}^{T-1} \gamma_t.$$

From (A4), there exists a real number $\bar{f}$ such that $\forall \boldsymbol{\theta} \in \Omega \Rightarrow f(\boldsymbol{\theta}) \leq \bar{f}$. Accordingly, we have

$$\Gamma_T = \frac{\mathbb{E}\left[f(\boldsymbol{\theta}_0)\right]}{\gamma_0} + \sum_{t=1}^{T-1} \left(\frac{\mathbb{E}\left[f(\boldsymbol{\theta}_t)\right]}{\gamma_t} - \frac{\mathbb{E}\left[f(\boldsymbol{\theta}_t)\right]}{\gamma_{t-1}}\right) - \frac{\mathbb{E}\left[f(\boldsymbol{\theta}_T)\right]}{\gamma_{T-1}}$$

$$= \frac{\mathbb{E}\left[f(\boldsymbol{\theta}_0)\right]}{\gamma_0} + \sum_{t=1}^{T-1} \left(\frac{1}{\gamma_t} - \frac{1}{\gamma_{t-1}}\right) \mathbb{E}\left[f(\boldsymbol{\theta}_t)\right] - \frac{\mathbb{E}\left[f(\boldsymbol{\theta}_T)\right]}{\gamma_{T-1}}$$

$$\leq \frac{\bar{f}}{\gamma_0} + \bar{f} \sum_{t=1}^{T-1} \left(\frac{1}{\gamma_t} - \frac{1}{\gamma_{t-1}}\right) - \frac{f(\boldsymbol{\theta}^\star)}{\gamma_{T-1}}$$

$$\leq \frac{\bar{f}}{\gamma_0} + \bar{f} \left(\frac{1}{\gamma_{T-1}} - \frac{1}{\gamma_0}\right) - \frac{f(\boldsymbol{\theta}^\star)}{\gamma_{T-1}}$$

$$= \frac{1}{\gamma_{T-1}} \left(\bar{f} - f(\boldsymbol{\theta}^\star)\right)$$

$$\leq \frac{1}{\gamma_{T-1}} \left(\bar{f} + |f(\boldsymbol{\theta}^\star)|\right)$$

$$\leq \frac{2}{\gamma_{T-1}} \max\{\bar{f}, |f(\boldsymbol{\theta}^\star)|\} \tag{13}$$

The first inequality follows from $\mathbb{E}\left[f(\boldsymbol{\theta}_T)\right] \geq f(\boldsymbol{\theta}^\star)$ and $\frac{1}{\gamma_t} - \frac{1}{\gamma_{t-1}} \geq 0$ since $\gamma_t$ is monotone decreasing. Hence,

$$\sum_{t=0}^{T-1} \mathbb{E}\left[\mathcal{G}(\boldsymbol{\theta}_t)\right] \leq \frac{2}{\gamma_{T-1}} \max\{\bar{f}, |f(\boldsymbol{\theta}^\star)|\} + \frac{D\sigma}{\sqrt{b}} T + \frac{LD^2}{2} \sum_{t=0}^{T-1} \gamma_t,$$

$$\frac{1}{T} \sum_{t=0}^{T-1} \mathbb{E}\left[\mathcal{G}(\boldsymbol{\theta}_t)\right] \leq \frac{2}{T\gamma_{T-1}} \max\{\bar{f}, |f(\boldsymbol{\theta}^\star)|\} + \frac{D\sigma}{\sqrt{b}} + \frac{LD^2}{2T} \sum_{t=0}^{T-1} \gamma_t.$$

*(i)* If we use $\gamma_t = \frac{1}{t+1}$, then

$$\gamma_{T-1} = \frac{1}{T} \text{ and } \frac{1}{T}\sum_{t=0}^{T-1}\gamma_t \leq \frac{1}{T}\left(1 + \int_0^{T-1}\frac{dt}{(t+1)}\right) \leq \frac{1+\log T}{T}.$$

Therefore,

$$\frac{1}{T}\sum_{t=0}^{T-1}\mathbb{E}\left[\mathcal{G}(\boldsymbol{\theta}_t)\right] \leq 2\max\{\bar{f}, |f(\boldsymbol{\theta}^\star)|\} + \frac{D\sigma}{\sqrt{b}} + \frac{LD^2(1+\log T)}{2T}$$

$$= \mathcal{O}\left(\frac{\log T}{T} + \frac{\sigma}{\sqrt{b}} + C\right),$$

where $C := 2\max\{\bar{f}, |f(\boldsymbol{\theta}^\star)|\}$.

*(ii)* If we use $\gamma_t = \frac{1}{(t+1)^a}(a \in [\frac{1}{2}, 1))$, then

$$\gamma_{T-1} = \frac{1}{T^a} \text{ and } \frac{1}{T}\sum_{t=0}^{T-1}\gamma_t \leq \frac{1}{T}\left(1 + \int_0^{T-1}\frac{dt}{(t+1)^a}\right) \leq \frac{1}{1-a}\cdot\frac{1}{T^a}.$$

Therefore,

$$\frac{1}{T}\sum_{t=0}^{T-1}\mathbb{E}\left[\mathcal{G}(\boldsymbol{\theta}_t)\right] \leq \frac{2\max\{\bar{f}, |f(\boldsymbol{\theta}^\star)|\}}{T^{1-a}} + \frac{D\sigma}{\sqrt{b}} + \frac{LD^2}{2(1-a)T^a}$$

$$= \mathcal{O}\left(\frac{1}{T^{\min\{1-a,a\}}} + \frac{\sigma}{\sqrt{b}}\right).$$

*(iii)* If we use $\gamma_t = (\underbrace{\gamma, \gamma, \cdots, \gamma}_{K}, \underbrace{\eta\gamma, \eta\gamma, \cdots, \eta\gamma}_{K}, \cdots, \underbrace{\eta^{P-1}\gamma, \eta^{P-1}\gamma, \cdots, \eta^{P-1}\gamma}_{K})$, then there exist real number $\underline{\gamma}$ such that $\gamma_{T-1} = \eta^{P-1}\gamma \geq \underline{\gamma}$. Hence, from Lemma A.5, we have that

$$\frac{1}{T}\sum_{t=0}^{T-1}\mathbb{E}\left[\mathcal{G}(\boldsymbol{\theta}_t)\right] \leq \frac{2\max\{\bar{f}, |f(\boldsymbol{\theta}^\star)|\}}{T\underline{\gamma}} + \frac{D\sigma}{\sqrt{b}} + \frac{LD^2}{2T}\frac{K\gamma}{1-\eta}$$

$$= \mathcal{O}\left(\frac{1}{T} + \frac{\sigma}{\sqrt{b}}\right).$$

This completes the proof. $\qquad\square$

## D   Proof of Theorem 3.4

*Proof.* From Lemma A.1, we have

$$\mathbb{E}\left[\mathcal{G}(\boldsymbol{\theta}_t)\right] \leq \frac{1}{\gamma_t}\left(\mathbb{E}\left[f(\boldsymbol{\theta}_t)\right] - \mathbb{E}\left[f(\boldsymbol{\theta}_{t+1})\right]\right) + \frac{D\sigma}{\sqrt{b_t}} + \frac{LD^2\gamma_t}{2}.$$

Summing over $t$, we get

$$\sum_{t=0}^{T-1}\mathbb{E}\left[\mathcal{G}(\boldsymbol{\theta}_t)\right] \leq \underbrace{\sum_{t=0}^{T-1}\frac{1}{\gamma_t}\left(\mathbb{E}\left[f(\boldsymbol{\theta}_t)\right] - \mathbb{E}\left[f(\boldsymbol{\theta}_{t+1})\right]\right)}_{\Gamma_T} + D\sigma\sum_{t=0}^{T-1}\frac{1}{\sqrt{b_t}} + \frac{LD^2}{2}\sum_{t=0}^{T-1}\gamma_t.$$

From Equation (13) and Lemma A.4, we have

$$\frac{1}{T}\sum_{t=0}^{T-1}\mathbb{E}\left[\mathcal{G}(\boldsymbol{\theta}_t)\right] \leq \frac{2}{T\gamma_{T-1}}\max\{\bar{f}, |f(\boldsymbol{\theta}^\star)|\} + \frac{D\sigma}{T}\frac{E\sqrt{\lambda}}{\sqrt{b}(\sqrt{\lambda}-1)} + \frac{LD^2}{2T}\sum_{t=0}^{T-1}\gamma_t.$$

*(i)* If we use $\gamma_t = \frac{1}{(t+1)}$, then

$$\gamma_{T-1} = \frac{1}{T} \text{ and } \frac{1}{T}\sum_{t=0}^{T-1}\gamma_t \leq \frac{1}{T}\left(1 + \int_0^{T-1}\frac{dt}{(t+1)}\right) \leq \frac{1 + \log T}{T}.$$

Therefore,

$$\frac{1}{T}\sum_{t=0}^{T-1}\mathbb{E}\left[\mathcal{G}(\boldsymbol{\theta}_t)\right] \leq 2\max\{\bar{f}, |f(\boldsymbol{\theta}^\star)|\} + \frac{D\sigma}{T}\frac{E\sqrt{\lambda}}{\sqrt{b}(\sqrt{\lambda}-1)} + \frac{LD^2(1 + \log T)}{2T}$$

$$= \mathcal{O}\left(\frac{\log T}{T} + C\right),$$

where $C := 2\max\{\bar{f}, |f(\boldsymbol{\theta}^\star)|\}$.

*(ii)* If we use $\gamma_t = \frac{1}{(t+1)^a}(a \in [\frac{1}{2}, 1))$, then

$$\gamma_{T-1} = \frac{1}{T^a} \text{ and } \frac{1}{T}\sum_{t=0}^{T-1}\gamma_t \leq 1 + \int_0^{T-1}\frac{dt}{(t+1)^a} \leq \frac{1}{1-a}\cdot\frac{1}{T^a}.$$

Therefore,

$$\frac{1}{T}\sum_{t=0}^{T-1}\mathbb{E}\left[\mathcal{G}(\boldsymbol{\theta}_t)\right] \leq \frac{2\max\{\bar{f}, |f(\boldsymbol{\theta}^\star)|\}}{T^{1-a}} + \frac{D\sigma}{T}\frac{E\sqrt{\lambda}}{\sqrt{b}(\sqrt{\lambda}-1)} + \frac{LD^2}{2(1-a)T^a}$$

$$= \mathcal{O}\left(\frac{1}{T^{\min\{1-a,a\}}}\right).$$

*(iii)* If we use $\gamma_t = (\underbrace{\gamma, \gamma, \cdots, \gamma}_{K}, \underbrace{\eta\gamma, \eta\gamma, \cdots, \eta\gamma}_{K}, \cdots, \underbrace{\eta^{P-1}\gamma, \eta^{P-1}\gamma, \cdots, \eta^{P-1}\gamma}_{K})$, then there exist real number $\underline{\gamma}$ such that $\gamma_{T-1} = \eta^{P-1}\gamma \geq \underline{\gamma}$ . Hence, from Lemma A.5, we have that

$$\frac{1}{T}\sum_{t=0}^{T-1}\mathbb{E}\left[\mathcal{G}(\boldsymbol{\theta}_t)\right] \leq \frac{2\max\{\bar{f}, |f(\boldsymbol{\theta}^\star)|\}}{T\underline{\gamma}} + \frac{D\sigma}{T}\frac{E\sqrt{\lambda}}{\sqrt{b}(\sqrt{\lambda}-1)} + \frac{LD^2}{2T}\frac{K\gamma}{1-\eta}$$

$$= \mathcal{O}\left(\frac{1}{T}\right).$$

This completes the proof. $\square$

# E   Full Experimental Results

The code used is available at our GitHub repository (`https://anonymous.4open.science/r/sfw25`). The experimental environment consisted of NVIDIA DGX A100×8GPU and Dual AMD Rome7742 2.25-GHz, 128 Cores×2CPU. The software environment was Python 3.8.2, Pytorch 1.6.0, and CUDA 11.6. All experiments were performed using a single GPU.

## E.1   Supplemental Results for Section 3.3

Figure 3 plots the test accuracy and loss function values for a batch size of $2^5$ and a constant learning rate $\gamma_t = \gamma_0 = 0.1$. Figure 4 similarly plots the test accuracy and loss function values for a batch size of $2^5$ and a decaying learning rate that halves every 40 epochs; i.e., this setting is based on Theorems 3.3(ii). Both figures show that SFW achieves higher test accuracy and lower loss function values than those of SFWM, in contrast to Figures 1 and 2.

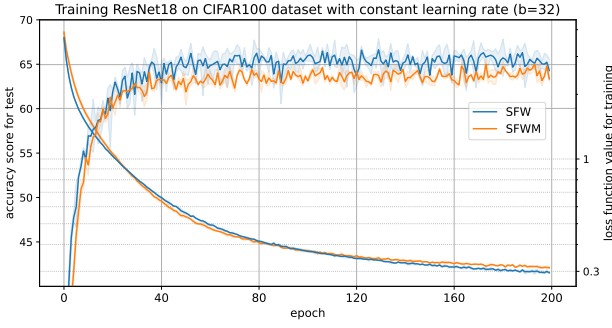

Figure 3: Accuracy score for the testing and loss function value for training versus the number of epochs in training ResNet18 on the CIFAR100 dataset with the $L_2$ constraint and a constant learning rate. The solid line represents the mean value, and the shaded area represents the maximum and minimum over three runs.

Figure 4: Accuracy score for the testing and loss function value for training versus the number of epochs in training ResNet18 on the CIFAR100 dataset with the $L_2$ constraint and a decaying learning rate. The solid line represents the mean value, and the shaded area represents the maximum and minimum over three runs. The learning rate was decreased every 40 epochs as $[0.1, 0.05, 0.025, 0.0125, 0.00625]$ and batch size was fixed at $2^5$.

We also performed similar experiments for decaying learning rates (I) $\gamma_t := \frac{1}{t+1}$ based on Theorems 3.3(i) and (II) $\gamma_t := \frac{1}{\sqrt{t+1}}$ based on Theorems 3.3(ii). Theoretically, this is an excellent learning rate setting that can remove the extra term in the upper bound of the Frank Wolfe gap, but experimentally, it is found to be unusable because the learning rate becomes too small from the early stages of learning (see Figures 5 and 6).

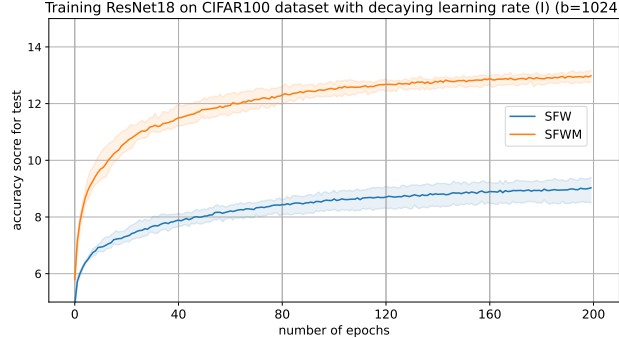

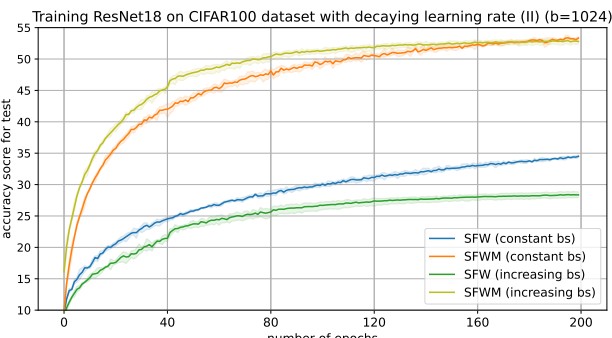

Figure 5: Accuracy score for the testing versus the number of epochs in training ResNet18 on the CIFAR100 dataset with the $L_2$ constraint and a decaying learning rate (I) $\gamma_t := \frac{1}{t+1}$. The batch size was fixed at $2^{10}$. The solid line represents the mean value, and the shaded area represents the maximum and minimum over three runs.

Figure 6: Accuracy score for the testing versus the number of epochs in training ResNet18 on the CIFAR100 dataset with the $L_2$ constraint and a decaying learning rate (II) $\gamma_t := \frac{1}{\sqrt{t+1}}$. The batch size was fixed at $2^{10}$. The solid line represents the mean value, and the shaded area represents the maximum and minimum over three runs.

### E.2   Supplemental Results for Section 4.5

We attacked the robust models listed in RobustBench (Croce et al., 2021) with PGD, FW, APGD, and AFW to verify their performance.

Table 5: Adversarial accuracy achieved by PGD and FW. CIFAR100 dataset/$L_\infty$ with $\epsilon = 8/255$

| paper | Architecture | clean accuracy | PGD | FW |
|---|---|---|---|---|
| (Wang et al., 2023) | WideResNet-70-16 | 75.22 | 48.41 | 48.44 |
| (Wang et al., 2023) | WideResNet-28-10 | 72.58 | 44.21 | 44.26 |
| (Debenedetti et al., 2023) | XCiT-L12 | 70.76 | 39.28 | 39.39 |
| (Rebuffi et al., 2021) | WideResNet-70-16 | 63.56 | 38.58 | 38.69 |
| (Debenedetti et al., 2023) | XCiT-M12 | 69.21 | 39.22 | 39.25 |
| (Pang et al., 2022) | WideResNet-70-16 | 65.56 | 36.87 | 36.94 |
| (Debenedetti et al., 2023) | XCiT-S12 | 67.34 | 37.47 | 37.42 |
| (Rebuffi et al., 2021) | WideResNet-28-16 | 62.41 | 36.09 | 36.13 |
| (Jia et al., 2022) | WideResNet-34-20 | 67.31 | 37.20 | 37.66 |
| (Addepalli et al., 2022a) | WideResNet-34-10 | 68.75 | 37.03 | 37.21 |
| (Cui et al., 2021) | WideResNet-34-10 | 62.97 | 37.48 | 37.8 |
| (Sehwag et al., 2022) | WideResNet-34-10 | 65.93 | 36.04 | 36.15 |
| (Pang et al., 2022) | WideResNet-28-10 | 63.66 | 35.39 | 35.43 |
| (Jia et al., 2022) | WideResNet-34-10 | 64.89 | 36.19 | 36.57 |
| (Addepalli et al., 2022b) | WideResNet-34-10 | 65.73 | 36.90 | 36.64 |
| (Cui et al., 2021) | WideResNet-34-20 | 62.55 | 34.62 | 34.63 |
| (Cui et al., 2021) | WideResNet-34-10 | 60.64 | 34.55 | 34.71 |
| (Rade & Moosavi-Dezfooli, 2021) | PreActResNet-18 | 61.50 | 32.69 | 32.75 |
| (Wu et al., 2020) | WideResNet-34-10 | 60.38 | 33.65 | 33.73 |
| (Rebuffi et al., 2021) | PreActResNet-18 | 56.87 | 31.95 | 32.0 |
| (Hendrycks et al., 2019) | WideResNet-28-10 | 59.23 | 33.75 | 33.79 |
| (Addepalli et al., 2022a) | ResNet-18 | 65.45 | 33.79 | 33.97 |
| (Cui et al., 2021) | WideResNet-34-10 | 70.25 | 30.41 | 30.51 |
| (Addepalli et al., 2022b) | PreActResNet-18 | 62.02 | 33.14 | 33.29 |
| (Chen et al., 2022) | WideResNet-34-10 | 62.15 | 31.53 | 31.87 |
| (Rice et al., 2020) | PreActResNet-18 | 53.83 | 20.95 | 21.01 |

Table 6: Adversarial accuracy achieved by PGD and FW. ImageNet dataset/$L_\infty$ with $\epsilon = 4/255$

| paper | Architecture | clean accuracy | PGD | FW |
|---|---|---|---|---|
| (Liu et al., 2023) | ConvNeXt-L | 78.02 | 60.66 | 60.68 |
| (Singh et al., 2023) | ConvNeXt-L + ConvStem | 77.00 | 59.26 | 59.26 |
| (Singh et al., 2023) | ConvNeXt-B + ConvStem | 75.88 | 58.56 | 58.60 |
| (Liu et al., 2023) | ConvNeXt-B | 76.7 | 58.30 | 58.32 |
| (Singh et al., 2023) | ViT-B + ConvStem | 76.30 | 57.12 | 57.18 |
| (Singh et al., 2023) | ConvNeXt-S + ConvStem | 74.08 | 55.22 | 55.24 |
| (Singh et al., 2023) | ConvNeXtt-T + ConvStem | 72.70 | 53.32 | 53.36 |
| (Singh et al., 2023) | ViT-S + ConvStem | 72.58 | 51.34 | 51.36 |
| (Debenedetti et al., 2023) | XCiT-L12 | 73.76 | 49.88 | 49.9 |
| (Debenedetti et al., 2023) | XCiT-M12 | 74.04 | 48.14 | 48.1 |
| (Debenedetti et al., 2023) | XCiT-S12 | 72.34 | 45.28 | 45.3 |
| (Salman et al., 2020) | WideResNet-50-2 | 68.64 | 41.42 | 41.60 |
| (Salman et al., 2020) | ResNet-50 | 64.06 | 39.18 | 39.28 |
| (Ilyas et al., 2019) | ResNet-50 | 62.52 | 33.24 | 33.40 |
| (Wong et al., 2020) | ResNet-50 | 55.64 | 30.50 | 30.12 |
| (Salman et al., 2020) | ResNet-18 | 52.92 | 29.98 | 30.14 |

Table 7: Adversarial accuracy achieved by PGD and FW. CIFAR10 dataset/$L_\infty$ with $\epsilon = 8/255$

| paper | Architecture | clean accuracy | PGD | FW |
|---|---|---|---|---|
| (Wang et al., 2023) | WideResNet-70-16 | 93.25 | 73.62 | 73.62 |
| (Wang et al., 2023) | WideResNet-28-10 | 92.44 | 70.39 | 70.38 |
| (Rebuffi et al., 2021) | WideResNet-70-16 | 92.23 | 69.86 | 69.99 |
| (Gowal et al., 2021) | WideResNet-70-16 | 88.74 | 68.97 | 69.05 |
| (Rebuffi et al., 2021) | WideResNet-106-16 | 88.50 | 68.07 | 68.26 |
| (Rebuffi et al., 2021) | WideResNet-70-16 | 88.54 | 67.65 | 67.84 |
| (Kang et al., 2021) | WideResNet-70-16 | 93.73 | 90.86 | 90.72 |
| (Xu et al., 2023) | WideResNet-28-10 | 93.69 | 67.21 | 67.13 |
| (Gowal et al., 2021) | WideResNet-28-10 | 87.50 | 65.94 | 66.09 |
| (Pang et al., 2022) | WideResNet-70-16 | 89.01 | 66.81 | 66.83 |
| (Rade & Moosavi-Dezfooli, 2021) | WideResNet-34-10 | 91.47 | 65.89 | 65.89 |
| (Sehwag et al., 2022) | ResNest152 | 87.30 | 65.25 | 65.28 |
| (Huang et al., 2021a) | WideResNet-34-R | 91.23 | 65.06 | 65.23 |
| (Huang et al., 2021a) | WideResNet-34-R | 90.56 | 64.12 | 64.42 |
| (Dai et al., 2022) | WideResNet-28-10-PSSiLU | 87.02 | 64.14 | 64.18 |
| (Pang et al., 2022) | WideResNet-28-10 | 88.61 | 64.87 | 64.86 |
| (Rade & Moosavi-Dezfooli, 2021) | WideResNet-28-10 | 88.16 | 64.87 | 63.89 |
| (Rebuffi et al., 2021) | WideResNet-28-10 | 87.33 | 64.22 | 64.39 |
| (Sridhar et al., 2022) | WideResNet-34-15 | 86.53 | 63.20 | 63.43 |
| (Wu et al., 2020) | WideResNet-28-10 | 88.25 | 63.58 | 63.7 |
| (Sridhar et al., 2022) | WideResNet-28-10 | 89.46 | 62.56 | 62.65 |
| (Zhang et al., 2021) | WideResNet-28-10 | 89.36 | 67.64 | 67.81 |
| (Carmon et al., 2019) | WideResNet-28-10 | 89.69 | 62.31 | 62.39 |
| (Gowal et al., 2021) | PreActResNet-18 | 87.35 | 61.13 | 61.14 |
| (Addepalli et al., 2022b) | WideResNet-34-10 | 85.32 | 64.86 | 65.06 |
| (Addepalli et al., 2022a) | WideResNet-34-10 | 88.71 | 61.10 | 61.16 |
| (Rade & Moosavi-Dezfooli, 2021) | PreActResNet-18 | 89.02 | 61.58 | 61.52 |
| (Jia et al., 2022) | WideResNet-70-16 | 85.66 | 60.96 | 61.18 |
| (Debenedetti et al., 2023) | XCiT-L12 | 91.73 | 59.18 | 59.31 |
| (Debenedetti et al., 2023) | XCiT-M12 | 91.30 | 59.09 | 59.26 |
| (Sehwag et al., 2020) | WideResNet-28-10 | 88.98 | 59.94 | 60.08 |
| (Rebuffi et al., 2021) | PreActResNet-18 | 83.53 | 59.66 | 59.82 |
| (Wang et al., 2020) | WideResNet-28-10 | 87.50 | 62.65 | 62.69 |
| (Jia et al., 2022) | WideResNet-34-10 | 84.98 | 59.79 | 60.04 |
| (Wu et al., 2020) | WideResNet-34-10 | 85.36 | 59.17 | 59.24 |
| (Debenedetti et al., 2023) | XCiT-S12 | 90.06 | 58.98 | 58.99 |
| (Sehwag et al., 2022) | ResNet-18 | 84.59 | 58.81 | 58.79 |
| (Hendrycks et al., 2019) | WideResNet-28-10 | 87.11 | 57.58 | 57.68 |
| (Pang et al., 2020) | WideResNet-34-20 | 85.14 | 62.19 | 62.37 |
| (Cui et al., 2021) | WideResNet-34-20 | 88.70 | 55.44 | 55.53 |
| (Zhang et al., 2020) | WideResNet-34-10 | 84.52 | 57.09 | 57.13 |
| (Rice et al., 2020) | WideResNet-34-20 | 85.34 | 57.32 | 57.33 |
| (Huang et al., 2020b) | WideResNet-34-10 | 83.48 | 56.15 | 56.20 |
| (Zhang et al., 2019b) | WideResNet-34-10 | 84.92 | 55.09 | 55.13 |
| (Cui et al., 2021) | WideResNet-34-10 | 88.22 | 54.24 | 54.35 |
| (Addepalli et al., 2022a) | ResNet-18 | 85.71 | 56.56 | 56.58 |
| (Chen et al., 2020b) | ResNet-50 | 86.04 | 54.34 | 54.37 |
| (Chen et al., 2022) | WideResNet-34-10 | 85.32 | 54.60 | 55.12 |
| (Addepalli et al., 2022b) | ResNet-18 | 80.24 | 56.19 | 56.29 |
| (Zhang et al., 2019a) | WideResNet-34-10 | 87.20 | 46.37 | 46.38 |
| (Andriushchenko & Flammarion, 2020) | PreActResNet-18 | 79.84 | 47.42 | 47.49 |
| (Wong et al., 2020) | PreActResNet-18 | 83.34 | 46.55 | 46.74 |
| (Ding et al., 2018) | WideResNet-28-4 | 84.36 | 51.18 | 51.43 |

Table 8: Adversarial accuracy achieved by APGD and AFW. CIFAR10 dataset/$L_\infty$ with $\epsilon = 8/255$

| paper | Architecture | clean accuracy | APGD | AFW |
|---|---|---|---|---|
| (Wang et al., 2023) | WideResNet-70-16 | 93.25 | 73.51 | 73.5 |
| (Wang et al., 2023) | WideResNet-28-10 | 92.44 | 70.23 | 70.32 |
| (Rebuffi et al., 2021) | WideResNet-70-16 | 92.23 | 69.4 | 69.54 |
| (Gowal et al., 2021) | WideResNet-70-16 | 88.74 | 68.52 | 68.56 |
| (Rebuffi et al., 2021) | WideResNet-106-16 | 88.50 | 67.64 | 67.82 |
| (Rebuffi et al., 2021) | WideResNet-70-16 | 88.54 | 67.27 | 67.37 |
| (Kang et al., 2021) | WideResNet-70-16 | 93.73 | 84.98 | 86.98 |
| (Xu et al., 2023) | WideResNet-28-10 | 93.69 | 67.05 | 67.16 |
| (Gowal et al., 2021) | WideResNet-28-10 | 87.50 | 65.56 | 65.77 |
| (Pang et al., 2022) | WideResNet-70-16 | 89.01 | 66.69 | 66.74 |
| (Rade & Moosavi-Dezfooli, 2021) | WideResNet-34-10 | 91.47 | 65.66 | 65.73 |
| (Sehwag et al., 2022) | ResNest152 | 87.30 | 65.08 | 65.06 |
| (Huang et al., 2021a) | WideResNet-34-R | 91.23 | 64.52 | 64.77 |
| (Huang et al., 2021a) | WideResNet-34-R | 90.56 | 63.55 | 63.8 |
| (Dai et al., 2022) | WideResNet-28-10-PSSiLU | 87.02 | 63.99 | 64.02 |
| (Pang et al., 2022) | WideResNet-28-10 | 88.61 | 64.73 | 64.73 |
| (Rade & Moosavi-Dezfooli, 2021) | WideResNet-28-10 | 88.16 | 63.78 | 63.81 |
| (Rebuffi et al., 2021) | WideResNet-28-10 | 87.33 | 63.99 | 64.08 |
| (Sridhar et al., 2022) | WideResNet-34-15 | 86.53 | 62.96 | 63.03 |
| (Wu et al., 2020) | WideResNet-28-10 | 88.25 | 63.35 | 63.49 |
| (Sridhar et al., 2022) | WideResNet-28-10 | 89.46 | 62.06 | 62.27 |
| (Zhang et al., 2021) | WideResNet-28-10 | 89.36 | 66.38 | 66.58 |
| (Carmon et al., 2019) | WideResNet-28-10 | 89.69 | 61.71 | 62.06 |
| (Gowal et al., 2021) | PreActResNet-18 | 87.35 | 60.84 | 60.94 |
| (Addepalli et al., 2022b) | WideResNet-34-10 | 85.32 | 64.41 | 64.52 |
| (Addepalli et al., 2022a) | WideResNet-34-10 | 88.71 | 60.79 | 60.96 |
| (Rade & Moosavi-Dezfooli, 2021) | PreActResNet-18 | 89.02 | 61.39 | 61.46 |
| (Jia et al., 2022) | WideResNet-70-16 | 85.66 | 60.63 | 60.61 |
| (Debenedetti et al., 2023) | XCiT-L12 | 91.73 | 58.98 | 59.18 |
| (Debenedetti et al., 2023) | XCiT-M12 | 91.30 | 59.01 | 59.03 |
| (Sehwag et al., 2020) | WideResNet-28-10 | 88.98 | 59.62 | 59.79 |
| (Rebuffi et al., 2021) | PreActResNet-18 | 83.53 | 59.58 | 59.64 |
| (Wang et al., 2020) | WideResNet-28-10 | 87.50 | 61.72 | 61.95 |
| (Jia et al., 2022) | WideResNet-34-10 | 84.98 | 59.51 | 59.58 |
| (Wu et al., 2020) | WideResNet-34-10 | 85.36 | 58.79 | 58.91 |
| (Debenedetti et al., 2023) | XCiT-S12 | 90.06 | 58.78 | 58.84 |
| (Sehwag et al., 2022) | ResNet-18 | 84.59 | 58.4 | 58.54 |
| (Hendrycks et al., 2019) | WideResNet-28-10 | 87.11 | 57.2 | 57.38 |
| (Pang et al., 2020) | WideResNet-34-20 | 85.14 | 61.59 | 61.78 |
| (Cui et al., 2021) | WideResNet-34-20 | 88.70 | 55.03 | 55.29 |
| (Zhang et al., 2020) | WideResNet-34-10 | 84.52 | 56.73 | 56.95 |
| (Rice et al., 2020) | WideResNet-34-20 | 85.34 | 56.89 | 57.04 |
| (Huang et al., 2020b) | WideResNet-34-10 | 83.48 | 55.77 | 55.99 |
| (Zhang et al., 2019b) | WideResNet-34-10 | 84.92 | 54.79 | 55.00 |
| (Cui et al., 2021) | WideResNet-34-10 | 88.22 | 53.84 | 54.11 |
| (Addepalli et al., 2022a) | ResNet-18 | 85.71 | 56.25 | 56.36 |
| (Chen et al., 2020b) | ResNet-50 | 86.04 | 54.22 | 54.29 |
| (Chen et al., 2022) | WideResNet-34-10 | 85.32 | 53.72 | 54.11 |
| (Addepalli et al., 2022b) | ResNet-18 | 80.24 | 55.84 | 55.97 |
| (Zhang et al., 2019a) | WideResNet-34-10 | 87.20 | 46.1 | 46.23 |
| (Andriushchenko & Flammarion, 2020) | PreActResNet-18 | 79.84 | 46.95 | 47.1 |
| (Wong et al., 2020) | PreActResNet-18 | 83.34 | 45.89 | 46.16 |
| (Ding et al., 2018) | WideResNet-28-4 | 84.36 | 50.14 | 50.33 |

Table 9: Adversarial accuracy achieved by APGD and AFW. ImageNet dataset/$L_\infty$ with $\epsilon = 4/255$

| paper | Architecture | clean accuracy | APGD | AFW |
|---|---|---|---|---|
| (Liu et al., 2023) | ConvNeXt-L | 78.02 | 60.28 | 60.34 |
| (Singh et al., 2023) | ConvNeXt-L + ConvStem | 77.00 | 58.98 | 59.02 |
| (Singh et al., 2023) | ConvNeXt-B + ConvStem | 75.88 | 58.36 | 58.4 |
| (Liu et al., 2023) | ConvNeXt-B | 76.7 | 58.02 | 58.06 |
| (Singh et al., 2023) | ViT-B + ConvStem | 76.30 | 56.92 | 57.0 |
| (Singh et al., 2023) | ConvNeXt-S + ConvStem | 74.08 | 55.00 | 55.06 |
| (Singh et al., 2023) | ConvNeXtt-T + ConvStem | 72.70 | 52.94 | 53.08 |
| (Singh et al., 2023) | ViT-S + ConvStem | 72.58 | 50.98 | 51.04 |
| (Debenedetti et al., 2023) | XCiT-L12 | 73.76 | 49.6 | 49.66 |
| (Debenedetti et al., 2023) | XCiT-M12 | 74.04 | 47.6 | 47.7 |
| (Debenedetti et al., 2023) | XCiT-S12 | 72.34 | 44.76 | 44.94 |
| (Salman et al., 2020) | WideResNet-50-2 | 68.64 | 40.86 | 40.9 |
| (Salman et al., 2020) | ResNet-50 | 64.06 | 38.50 | 38.76 |
| (Ilyas et al., 2019) | ResNet-50 | 62.52 | 32.18 | 32.42 |
| (Wong et al., 2020) | ResNet-50 | 55.64 | 29.38 | 28.54 |
| (Salman et al., 2020) | ResNet-18 | 52.92 | 29.26 | 29.48 |

