# OpenReview forum: "Stochastic Frank Wolfe for Constrained Nonconvex Optimization"
_TMLR — Rejected by TMLR_

### Review · Reviewer_fFgk · 2025-02-23

**Summary Of Contributions:**

This paper investigates stochastic frank wolfe (SFW) methods for constrained nonconvex optimization. The authors set decaying learning rates and increasing bacth size for SFW methods, which ensures that the FW gap converges to zero in practical analysis. Furthermore, they also apply SFW algorithms to adversarial attacks and propose a new adversarial attack method. Finally, they conduct experiments to validate the effectiveness of their methods.

**Audience:**

Yes

**Broader Impact Concerns:**

This work is mainly theoretical.

**Claims And Evidence:**

No

**Requested Changes:**

* The authors have overlooked a related work on SFW methods. Yurtsever, A., Sra, S., and Cevher, V. Conditional gradient methods via stochastic path-integrated differential estimator. ICML, 2019.
* It is unnecessary to include "(ours)" in Table 1. Table 2 already clearly summarizes the contributions of this paper. Additionally, in Table 1, the term "increasing" with respect to the Momentum of the SCG algorithm is not explained.
* The authors should include a comparison of the convergence rates of other SFW algorithms with the theoretical results of this paper, as this would better illustrate the motivation behind their work.

**Strengths And Weaknesses:**

The content of this paper is substantial, and its structure is clear. In addition, this paper is well-written. The motivation behind this work is reasonable, as the authors aim to demonstrate that the learning rates set by most existing SFW methods are predetermined and fixe, whereas time-varying learning rates are more suitable.

My main concern lies with the theoretical results of this work. The technical contribution is the proposed new setting of decaying learning rate and increasing batch size  in Eq. (1) and (2), where they require $PK=T$ and $QE=T$. In Theorem 3.5, the authors establish the convergence rate with respect to $E$ and $\gamma_{\infty}$. However, $E$ depends on $T$, and the authors do not provide a specific upper bound for $\gamma_{\infty}$. In my view, stating only that $\gamma_{\infty}<+\infty$  is insufficient, as the upper bound for $\gamma_{\infty}$ should depend on $P$ and $K$. Therefore, it remains unclear whether the dependence of $\gamma_{\infty}$ on $T$ will affect the optimality of the convergence rate. Additionally, Theorem 3.5 (and other related theorems) does not clearly specify the settings for $\eta$ and $\gamma$.

---

> ### Author Response · Authors · 2025-03-14
>
> Thank you for your thoughtful review. Please find the revised paper and our responses below. If you have any further concerns, please let us know.
>
> ---
>
> **Reply to Weakness:** To address your point, we have added Lemma A.5 and eliminated $\gamma_\infty$ from our theorem. Also, for $PK=T$ and $QE=T$, we have fixed the number of steps $K$ and $E$ in each stage, so they do not depend on $T$ . We have clarified this point in the revised manuscript. If you still have concerns, please reply.
>
> ---
>
> **Requested Change 1:** The authors have overlooked a related work on SFW methods. Yurtsever, A., Sra, S., and Cevher, V. Conditional gradient methods via stochastic path-integrated differential estimator. ICML, 2019.
>
> **Reply:** We have additionally cited it appropriately. Thank you for sharing this literature with us.
>
> ---
>
> **Requested Change 2:** It is unnecessary to include "(ours)" in Table 1. Table 2 already clearly summarizes the contributions of this paper. Additionally, in Table 1, the term "increasing" with respect to the Momentum of the SCG algorithm is not explained.
>
> **Reply:** Thanks for the helpful advice. We have removed our results from Table 1. We also removed the Mokhtari's SCG with increasing momentum since it was a result for convex optimization. We thank you for your valid point.
>
> ---
>
> **Requested Change 3:** The authors should include a comparison of the convergence rates of other SFW algorithms with the theoretical results of this paper, as this would better illustrate the motivation behind their work.
>
> **Reply:** We agree with you. To avoid redundancy, we have revised Table 1 to summarize the convergence rates from previous studies. We hope this modification meets your expectations.

---

> > ### Author Response · Authors · 2025-03-27
> >
> > Dear Reviewer fFgk,
> >
> > We sincerely appreciate your time and feedback. We have carefully addressed the reviewers' comments and revised the manuscript accordingly. If you have any remaining questions or concerns, please do not hesitate to let us know. We would be happy to address them. Thank you for your time and consideration.

---

### Review · Reviewer_8w4Z · 2025-02-27

**Summary Of Contributions:**

This contributions of the paper are a convergence analysis of the Stochastic Frank-Wolfe (SFW) and SFW with momentum (SFWM) methods for minimizing a finite-sum of Lipschitz-smooth nonconvex functions over a convex, compact constraint set.

The paper analyzes these algorithms under a few learning rate schedules (constant + a few decaying versions) and batch size strategies (constant and increasing). It demonstrates theoretically that the Frank-Wolfe gap converges to zero when using both decreasing learning rates and increasing batch sizes. It also a provides a new bound on the variance of the estimator introduced for Frank-Wolfe in Mokhtari et al 2018 when the momentum is constant.

The paper also applies a modification of the SFW algorithm to adversarial attacks on trained neural networks, proposing a new attack method called Auto-SFW (ASFW) which is analogous to Auto-Projected Gradient Descent (APGD) but without having to use projections.

---

"Stochastic conditional gradient methods: From convex minimization to submodular maximization" A Mokhtari, H Hassani, A Karbasi  - 2018

**Audience:**

Yes

**Broader Impact Concerns:**

I do not see any concerns on the ethical implications of the work that would require adding a Broader Impact Statement.

**Claims And Evidence:**

No

**Requested Changes:**

# Title

The title of this paper is **extremely** similar to "Stochastic Frank-Wolfe Methods for Nonconvex Optimization" S Reddi, S Sra, B Poczos, A Smola - 2016. I strongly suggest an original, more descriptive title than the one currently used.

# Section 1

## Impractical analyses
The paragraphs on "Impractical analyses" in Section 1.2 should be heavily edited or removed entirely.

What's written here does not reflect how convergence rates are actually used in practice; nobody runs an algorithm for $T=+\infty$ (for obvious reasons). Instead, one specifies some $\epsilon$ tolerance for our problem in advance, then one can fix the horizon $T$ (and subsequently the step size/other parameters) to reach that tolerance. There is no contradiction or impracticality of these results arising from the fact that one cannot take $T=+\infty$ as the purpose of these analyses is to understand how quickly the algorithm converges as a function of the desired tolerance. The claim that the gap is not guaranteed to converge to $0$ for a fixed $T$ is irrelevant; one can find an $\epsilon$-critical point for arbitrary $\epsilon>0$.

Also, most of the analyses given in this paper are themselves disconnected from being practical (except for the constant step size, constant batch size); the increasing batch size strategy is not reflective of what is done in practice with a critical batch size and neither are the decaying step size strategies.

# Section 2

- In the introduction it is written "The Frank Wolfe algorithm is a classical first-order method for solving convex optimization problems with
closed convex constraint sets." This is not true, it is necessary to have compact sets for the Frank-Wolfe algorithm. One cannot use the Frank-Wolfe algorithm over an arbitrary closed subspace, for example.

- In Problem 2.1 it is written "we would like to find a local minimizer of $f$ over $\Omega$" but this is not what is written below, where one finds an explicit minimization of $f$ over $\Omega$ (global minimization).

- Also in Problem 2.1, in view of the previous discussion about $T\to+\infty$, it is not meaningful to consider a finite-sum problem if one is going to have batch size that is increasing and then take $T\to+\infty$, as eventually you are computing a deterministic gradient and the problem is no longer stochastic.

- Regarding the algorithm attribution in the paper, there appears to be some inconsistency in how the momentum-based method is referenced. The paper attributes the naming convention to Pokutta et al. 2020 with the statement, "In accordance with Pokutta et al. (2020), we will refer to Algorithm 3 as SFW with momentum (SFWM)." However, Mokhtari et al. 2018 originally introduced this method with explicit reference to the momentum term in their paper, as shown in this quote "In this paper, we developed stochastic conditional gradient methods for solving convex minimization and submodular maximization problems. The main idea of the proposed methods in both domains was using a **momentum term** in the stochastic gradient approximation step to reduce the noise of the stochastic approximation." (emphasis my own) and in this quote from a later paper, Mokhtari et al 2019, "As we mentioned earlier, among the stochastic variants of FW, the **momentum stochastic Frank-Wolfe method proposed in [Mokhtari et al., 2018a,b]** is the only method that requires only one sample per iteration." (emphasis my own). For clarity and proper attribution of scientific contributions, it would be helpful to acknowledge Mokhtari et al. 2018 as the originators of the momentum-based approach/SFWM. This would also help readers better understand the relationship between what is called SFWM in this paper and what is called Mokhtari's SFW, which are mentioned separately in the experimental section, despite referring to fundamentally the same algorithmic approach with different assumptions and hyperparameter choices.

# Section 3

## Section 3.1 & 3.2

- It is not clear why in Theorem 3.1 the convergence rate is written using sum of the step sizes but in Theorem 3.2 and Theorem 3.3 there is an explicit upper bound for the sum of step sizes used instead; this should be made uniform if possible. Does the same reasoning used for those theorems not hold anymore? I would have expected a rate like O(1/log(T) + 1/sqrt(b)).

- In Theorem 3.4 and Theorem 3.5 it is claimed that $\theta^\star$ is just a local minimizer but this is not sufficient for the proofs, which require $\theta^\star\in\mathrm{argmin}_{\theta\in\Omega}f(\theta)$ (See changes requested for the proofs in the appendix).

## Section 3.3

- The use of an $L^2$-ball constraint on the weights of the network is not motivated at all in this paper. Indeed it is eventually admitted that, "In fact, the ResNet18 training we have performed in this section does not need to be constrained, and unconstrained optimization algorithms such as SGD are sufficient." which leads one to wonder why this problem was chosen in the first place since it does not match the theory and does not lead to practical benefits. If the purpose of this is to show that the theory is holding (and this is implicitly invoked when making claims about the empirical results being supported by theoretical findings), then the chosen problem should not have an objective which is not Lipschitz-smooth (resnet18 does not have smooth activation functions and does not adhere to the assumptions in your paper) and for which the constraint is contrived. On the other hand, if the purpose is to show some practical benefit then there must be a comparison to the standard optimizers for this problem, e.g., ADAM/ADAMW.

- The conclusions drawn at the end of this section regarding the relationship between momentum and batch size are possibly overstated. The momentum needs to decay to decrease the variance of the stochastic gradient estimator, this is known since the paper of Mokhtari in 2018 where it was used in the proof of convergence for the convex version of this problem. It is also confirmed for nonconvex problems in Pethick et al 2025, which also studied SFWM and should be cited. So, when it is written "Thus, for SFW, adding momentum helps when the batch size is large, but has the opposite effect when the batch size is small." these experiments do not justify this claim as a general trend; at the very least it should be added that these claims are for **fixed** momentum only.

- The claim "Furthermore, Figures 1 and 2 show that, for SFW, the increasing batch size achieves higher test accuracy and lower loss function values than the constant batch size does. This is also theoretically supported by Theorems 3.1,3.2,3.4(ii), and 3.5(ii)." is not entirely true, those theorems do not apply to this problem because this problem is not Lipschitz-smooth.

- The claim "There are not many constrained nonconvex optimization problems in the machine-learning field, and applications of SFW methods to them may be limited" is not true. Sparse or low-rank matrix factorization, reinforcement learning, bilevel problems, semi-supervised learning with manifold constraints, etc all provide examples of constrained nonconvex optimization problems.

- When it is claimed that the rising curve is for test accuracy in the figures, it should be explicitly stated whether this is for top-1 or top-5.

- There is no information about the effect of some crucial hyperparameter choices made (radius of the constraint set, momentum parameter value) in the paper.

# Section 4

- The motivation for this section should be more clearly presented. One sees in the table that APGD and ASFW perform roughly the same, with APGD often beating it slightly. APGD should have the same cost to run in this context as ASFW. If there is some benefit or motivation to using ASFW for this problem then it is not being communicated in this section.

- It is claimed "So in this paper, we evaluate the attack performance of SFW algorithms by attacking robust models trained with adversarial training and compare it with PGD." but I do not see any column where SFW is used to do the attack? If ASFW is what was meant then this should be written explicitly.

# Section 5

- The claim "In our analysis, the learning rate and the batch size are independent of unknown parameters and are experimentally realistic." is not correct. Increasing the batch size again and again is not experimentally realistic at all.

- The claim "We showed that the Frank Wolfe gap has a convergence rate of O(1/T ) only when we decrease the learning rate and increase the batch size." is not correct; it must be explicitly stated that this is for constant momentum. This can be overcome with momentum that goes to 0, e.g., as is done in Mokhtari et al 2018 and Pethick et al 2025.

# Appendix

- There are some results that are not novel e.g. Proposition A.1 and Lemma A.2. and it must be stated explicitly which results are being claimed as novel and which are not.

- In the proof of Theorem 3.1 and Theorem 3.2 after summing over $t=0$ to $t=T-1$ there is a lower bounding of $\mathbb{E}[f(\theta_T)]$ by $\mathbb{E}[f(\theta^\star)]$ which is only valid if $\theta^{\star}\in\mathrm{argmin}_{\theta\in\Omega}f(\theta)$ (this is related to the erroneous statement of Theorem 3.1 and Theorem 3.2). This also recurs in the proof of Theorem 3.3.

- There is an error as well in the proof of Theorem 3.4. It is claimed that $$1 + \int\limits_{0}^{T-1}\frac{dt}{(t+1)^a} \leq \frac{1}{1-a}\frac{1}{T^a}$$ but this is not true as stated, taking $T=2$ and $a=1/2$ one finds that the left hand side is $\sqrt{2}+(\sqrt{2}-1)$ and the right hand side is $\sqrt{2}$. Probably one should divide this quantity by $T$.

---

## Formatting corrections

### Section 2

- The first sentence immediately following the statement of the algorithms in Section 2.1, "The Frank Wolfe algorithm (Frank & Wolfe, 1956) (Algorithm 1) is a classical first-order method." is actually repeated from the introduction.

- Throughout the paper, sequences are not written consistently. E.g., in the contribution section it is written "Let $(\theta_t)\in\mathbb{R}^d$ be the sequence..." Later on it is written $(\theta_k)_{k=0}^t\subset \mathbb{R}$ so it is not clear what is intended. There is also inconsistency in the bolding of many quantities (e.g., $\xi_t$).

- When discussing the gap in this section you suppose that $\theta^\star$ is the local minimizer of $f$ over $\Omega$ but there can be many local minimizers and there can also be points for which the gap is $0$ which are NOT local minimizers and this should not be taken for granted; this algorithm does not guarantee to find a local minimizer.

### Section 3

- In the notations it is written that expectations are with respect to the randomness of $\xi$ but then in Theorem 3.1, Theorem 3.2, Theorem 3.3 we have the expectation of $f(\theta_0)$ and of $f(\theta^\star)$ which are deterministic quantities with respect to the randomness of $\xi$.

- In the statements of Theorem 3.4 and Theorem 3.5 it should be $(\gamma_t)_{t\in\mathbb{N}}$.

- Right before Section 3.3, one sees $(\theta_t)$ again for a sequence, which is inconsistent with the formatting in other parts of the paper.

---

"Training Deep Learning Models with Norm-Constrained LMOs"  T Pethick, W Xie, K Antonakopoulos, Z Zhu, A Silveti-Falls, V Cevher - 2025

"Deep neural network training with frank-wolfe" S Pokutta, C Spiegel, M Zimmer - 2020

"One sample stochastic frank-wolfe" M Zhang, Z Shen, A Mokhtari, H Hassani, A Karbasi - 2019

**Strengths And Weaknesses:**

# Strengths

- The paper addresses an important topic in optimization for machine learning by providing convergence analysis for the stochastic Frank-Wolfe and stochastic Frank-Wolfe with momentum methods for minimizing a Lipschitz-smooth nonconvex function over a compact, convex set.

- The Auto-SFW attack method proposed for adversarial attacks appears to perform comparably the analogous first-order algorithm using project, APGD.

# Weaknesses

- Several theoretical claims and analyses contain imprecisions or inaccuracies.

- The motivation section on "Impractical analyses" in Section 1.2 misrepresents how convergence rates are used in practice. Existing results and related work are also sometimes micharacterized.

- The experimental results in Section 3.3 lack necessary comparisons to standard optimizers (ADAM/ADAMW) for this setting and fail to motivate why L2-constrained ResNet18 was chosen in the first place.

- The ASFW algorithm proposed for adversarial attacks is disconnected from the rest of the paper and its theoretical analyses, having no convergence guarantees or discussion in this direction.

---

> ### Author Response · Authors · 2025-03-14
>
> Thank you for your thoughtful review. Please find the revised paper and our responses below. If you have any further concerns, please let us know.
>
> ---
>
> **Requested Change Comment 1:** The title of this paper is extremely similar to "Stochastic Frank-Wolfe Methods for Nonconvex Optimization" S Reddi, S Sra, B Poczos, A Smola - 2016. I strongly suggest an original, more descriptive title than the one currently used.
>
> **Reply to Requested Change Comment 1:** We can change the title to "Stochastic Frank Wolfe for Constrained Nonconvex Optimization: Application to Adversarial Attack". We hope this addresses your concern, but we are open to further suggestions if the reviewers or the action editor prefer a different title.
>
> ---
>
> **Requested Change Comment 2:** The paragraphs on "Impractical analyses" in Section 1.2 should be heavily edited or removed entirely. What's written here does not reflect how convergence rates are actually used in practice; nobody runs an algorithm for $T=+\infty$ (for obvious reasons). Instead, one specifies some $\epsilon$ tolerance for our problem in advance, then one can fix the horizon $T$ (and subsequently the step size/other parameters) to reach that tolerance. There is no contradiction or impracticality of these results arising from the fact that one cannot take $T=+\infty$ as the purpose of these analyses is to understand how quickly the algorithm converges as a function of the desired tolerance. The claim that the gap is not guaranteed to converge to 0 for a fixed $T$ is irrelevant; one can find an $\epsilon$-critical point for arbitrary $\epsilon > 0$.
>
> **Reply to Requested Change Comment 2:** We do not believe this statement is incorrect and have no plans to correct it.
> If you see a flaw in our logical explanation, please point it out specifically, as you have done elsewhere.
> Convergence of sequence $(a_n)$ to 0 is a necessary and sufficient condition of the following: $\forall \epsilon > 0, \exists n_0 \in \mathbb{N}, \forall n \geq n_0 \rightarrow |a_n| < \epsilon$. This is all we claim.
> The claim that ``one can find an $\epsilon$-critical point for arbitrary $\epsilon > 0$'' is false, because any $\epsilon$ is initially determined. In other words, one cannot adjust $T$ to match the determined $\epsilon$.
> You are right, this argument may bring a casual understanding of convergence. However, they should be treated with caution in that they make it appear as if the nonconvergent sequence converges to 0 on the order of $1/\sqrt{T}$, and should not be treated as a rigorous convergence analysis in the least.
>
> ---
>
> **Requested Change Comment 3:** Also, most of the analyses given in this paper are themselves disconnected from being practical (except for the constant step size, constant batch size); the increasing batch size strategy is not reflective of what is done in practice with a critical batch size and neither are the decaying step size strategies.
>
> **Reply to Requested Change Comment 3:** Increasing batch size may indeed be suboptimal in terms of computational complexity, but it is clear from our results and many previous studies that increasing batch size yields superior performance for the amount of computation allowed in practice, so the claim that it is impractical is false. The same is true for the decaying learning rate.
>
> ---
>
> **Requested Change Comment 4:** Also in Problem 2.1, in view of the previous discussion about $T \to + \infty$, it is not meaningful to consider a finite-sum problem if one is going to have batch size that is increasing and then take $T \to + \infty$, as eventually you are computing a deterministic gradient and the problem is no longer stochastic.
>
> **Reply to Requested Change Comment 4:** We do not specify which probability distribution the random variable $\boldsymbol{\xi}$ follows, and we consider arbitrary probability distributions; if $\boldsymbol{\xi}$ were uniformly distributed, even a full batch $(b=n)$ would not impair stochasticity, since the same data could be selected $n$ times.

---

> ### Author Response · Authors · 2025-03-14
>
> **Requested Change Comment 5:** Regarding the algorithm attribution in the paper, there appears to be some inconsistency in how the momentum-based method is referenced. The paper attributes the naming convention to Pokutta et al. 2020 with the statement, "In accordance with Pokutta et al. (2020), we will refer to Algorithm 3 as SFW with momentum (SFWM)." However, Mokhtari et al. 2018 originally introduced this method with explicit reference to the momentum term in their paper, as shown in this quote "In this paper, we developed stochastic conditional gradient methods for solving convex minimization and submodular maximization problems. The main idea of the proposed methods in both domains was using a momentum term in the stochastic gradient approximation step to reduce the noise of the stochastic approximation." (emphasis my own) and in this quote from a later paper, Mokhtari et al 2019, "As we mentioned earlier, among the stochastic variants of FW, the momentum stochastic Frank-Wolfe method proposed in [Mokhtari et al., 2018a,b] is the only method that requires only one sample per iteration." (emphasis my own).
>
> **Reply to Requested Change Comment 5:** We acknowledge Mokhtari et al. (2018) as the originator of the momentum-based approach/SFWM. As evidence of this, we have properly cited this prior work, and our citations and descriptions will naturally lead the reader to see that Mokhtari et al. (2018) is the originator of the momentum method.
> Given that the terms SGD and SGD with momentum are widely used in the machine learning field, the name SFW with momentum (SFWM) would be the most intuitive.
> Mokhtari et al. (2018) and other previous studies of methods that add momentum to SFW differ in the way they take the momentum term, and all use a different name than SFWM.
> Since Pokutta et al. (2020) was the first to explicitly state SFWM, we stated ``In accordance with Pokutta et al. (2020) ... ''. There is no inconsistency or problem with this statement.
>
> ---
>
> **Requested Change Comment 6:** It is not clear why in Theorem 3.1 the convergence rate is written using sum of the step sizes but in Theorem 3.2 and Theorem 3.3 there is an explicit upper bound for the sum of step sizes used instead; this should be made uniform if possible. Does the same reasoning used for those theorems not hold anymore? I would have expected a rate like O(1/log(T) + 1/sqrt(b)).
>
> **Reply to Requested Change Comment 6:** In Theorem 3.1, the convergence rate is not written using sum of the step size. We do not seem to understand your point. If our revised manuscript still needs further correction, please explain further.
>
> ---
>
> **Requested Change Comment 7:** The use of an $L_2$-ball constraint on the weights of the network is not motivated at all in this paper. Indeed it is eventually admitted that, "In fact, the ResNet18 training we have performed in this section does not need to be constrained, and unconstrained optimization algorithms such as SGD are sufficient." which leads one to wonder why this problem was chosen in the first place since it does not match the theory and does not lead to practical benefits. If the purpose of this is to show that the theory is holding (and this is implicitly invoked when making claims about the empirical results being supported by theoretical findings), then the chosen problem should not have an objective which is not Lipschitz-smooth (resnet18 does not have smooth activation functions and does not adhere to the assumptions in your paper) and for which the constraint is contrived. On the other hand, if the purpose is to show some practical benefit then there must be a comparison to the standard optimizers for this problem, e.g., ADAM/ADAMW.
>
> **Reply to Requested Change Comment 7:** Thank you very much for your opinion. Since we would like to focus on SFW in training DNN models, the experiments considered in Section 3 were general image classification tasks. We have added Adam's results.

---

> ### Author Response · Authors · 2025-03-14
>
> **Requested Change Comment 8:** The conclusions drawn at the end of this section regarding the relationship between momentum and batch size are possibly overstated. The momentum needs to decay to decrease the variance of the stochastic gradient estimator, this is known since the paper of Mokhtari in 2018 where it was used in the proof of convergence for the convex version of this problem. It is also confirmed for nonconvex problems in Pethick et al 2025, which also studied SFWM and should be cited. So, when it is written "Thus, for SFW, adding momentum helps when the batch size is large, but has the opposite effect when the batch size is small." these experiments do not justify this claim as a general trend; at the very least it should be added that these claims are for fixed momentum only.
>
> **Reply to Requested Change Comment 8:** You are correct, our discussion only considers fixed momentum. We have clearly stated this. We appreciate the suggestion. However, given that the paper was posted on arXiv just two days before our submission, we hope the expectation to cite it is not too strict.
>
> ---
>
> **Requested Change Comment 9:** The motivation for this section should be more clearly presented. One sees in the table that APGD and ASFW perform roughly the same, with APGD often beating it slightly. APGD should have the same cost to run in this context as ASFW. If there is some benefit or motivation to using ASFW for this problem then it is not being communicated in this section.
>
> **Reply to Requested Change Comment 9:** Our motivation was to see if the Frank Wolfe attack also works for robust models, as described in Section 1.2.
> In Section 4 we were able to confirm that it works, but not its benefit for APGD.
>
> ---
>
> **Requested Change Comment 10:** The claim "In our analysis, the learning rate and the batch size are independent of unknown parameters and are experimentally realistic." is not correct. Increasing the batch size again and again is not experimentally realistic at all.
>
> **Reply to Requested Change Comment 10:** A technique for increasing the batch size within a practically feasible computational budget is widely used. Our analysis theoretically follows this technique, so the claim that our approach is not experimentally realistic is not valid.
>
> ---
>
> **Requested Change Comment 11:** There is an error as well in the proof of Theorem 3.4. It is claimed that $1 + \int_{0}^{T-1}\frac{dt}{(t+1)^a} \leq \frac{1}{1-a}\frac{1}{T^a}$ but this is not true as stated, taking $T=2$ and $a=1/2$ one finds that the left hand side is $2+(2-1)$ and the right hand side is $2$. Probably one should divide this quantity by $T$.
>
> **Reply to Requested Change Comment 11:** This is the core of the proof of our theorem. Thank you for pointing out the typographical error.
>
> ---
>
> **Reply to Other Comments:** Thank you for your useful comments. Our paper has been revised appropriately.

---

> > ### Comment · Reviewer_8w4Z · 2025-03-24
> >
> > Thank you for taking into account my suggestions.
> >
> > ---
> >
> > ### Comment 2
> >
> > Your characterization of convergence of a sequence of real numbers is totally correct and I completely agree that these analyses do not show the convergence of the quantity of interest to 0. However, this does not preclude finding an $\epsilon$-critical point and it doesn’t mean these cannot be called convergence rates.
> >
> > One must specify $\epsilon$ before one runs the algorithm. This choice of $\epsilon$ induces a choice of $T$. Then, one runs the algorithm for the number of iterations $T$, with a stepsize that depends on $T$, to make the bound smaller than $\epsilon$. There is nothing casual about it - it’s mathematically rigorous to claim that this leads to an $\epsilon$-critical point.
> >
> > A simple example is if you have some quantity of interest $g_t$ which satisfies $g_T \leq \frac{1}{T\gamma} + \gamma$. Then taking $\gamma = \frac{1}{\sqrt{T}}$ would give a bound $g_T = \frac{2}{\sqrt{T}}$. To find an $\epsilon$-critical point, you would need $\frac{2}{\sqrt{T}}\leq \epsilon$ which is satisfied once $T \geq \frac{4}{\epsilon^2}$.
> >
> > “However, they should be treated with caution in that they make it appear as if the nonconvergent sequence converges to 0 on the order of , and should not be treated as a rigorous convergence analysis in the least.”
> >
> > This characterization is not correct. Papers studying algorithms with stepsizes that depend on $T$ (such as Reddi et al. 2016, Cutkosky et al 2020, etc) are not claiming to produce a sequence converging to 0, they produce a specific output $x_T$ guaranteed to be an $\epsilon$-critical point. The convergence rate here simply tells one how many iterations are required to guarantee a given accuracy, which is the standard nomenclature in the optimization community.
> >
> > "Stochastic Frank-Wolfe Methods for Nonconvex Optimization" Reddi, Sra, Poczos, Smola 2016
> > "Momentum Improves Normalized SGD" Cutkosky, Mehta 2020
> >
> > ### Comment 5
> >
> > The inconsistency comes from the fact that you are also calling it Mokhtari’s SFW in the plots when Mokhtari’s SFW is SFWM.
> >
> > ### Comment 6
> >
> > I referred to the wrong theorem, I should have written Theorem 3.3 when the stepsize is $\frac{1}{t+1}$.
> >
> > ### Comment 7
> >
> > If the focus is on training DNNs then comparing to untuned baselines is not informative and does not give an accurate presentation of which algorithms are performant.
> >
> > ### Comment 8
> >
> > The expectation is not based on when the paper was written but rather about presenting the whole picture when discussing the behavior of this algorithm and the effect of momentum.
> >
> > ### Comment 9
> >
> > I stand by my statement that this part of the paper is unmotivated and completely disconnected from the rest. It is not using SFWM (neither the stochasticity nor the momentum), it has no convergence guarantees or discussion in this direction, and it does not show any practical benefits over APGD. Although several prior works that studied Frank-Wolfe for adversarial attacks are cited (in a single sentence), they are not actually discussed, instead we just get "Note that Chen et al. (2020a); Sahu& Kar (2020); Huang et al. (2020a); Kazemi et al. (2021); Imtiaz et al. (2022) tackled adversarial attacks using the Frank Wolfe algorithm or its variants." without any substantial explanation of how this work relates to them. For example, in Chen et al 2020 the "Frank-Wolfe White-box Attack Algorithm" is actually SFWM without stochasticity, for adversarial attacks, but this is never mentioned in the current manuscript.
> >
> > ---
> >
> > #### Small errors
> >
> > Several times in the paper there is reference to using SFW for adversarial attacks, e.g.,
> >
> > “In this section, we explain that the optimization problem for the adversarial attack is an instance of a constrained non-convex optimization problem and experimentally verify whether this attack succeeds with SFW methods.”
> >
> > but this should just be FW now.
> >
> > “In this paper, SFW is used to attack”...
> >
> > “We attacked the robust models listed in RobustBench (Croce et al., 2021) with PGD, SFW, APGD, and AFW to verify their performance.”
> >
> > “We also showed experimentally that SFW algorithms perform as well as PGD in adversarial attacks and that our proposed AFW performs as well as APGD.”

---

> > > ### Author Response · Authors · 2025-03-24
> > >
> > > We deeply appreciate your reply. Please find the revised paper and our responses below. If you have any further concerns, please let us know. Please do not hesitate to reply if you are not satisfied, especially with reply to comment 2.
> > >
> > > ---
> > >
> > > **Reply to Comment 2:** We disagree with your assertion. In your example, you are correct that the choice of $\epsilon$ induces the choice of $T$, since $T \geq 4 /\epsilon^2$. We see that if we take $\epsilon$ small, we need a larger $T$. But this is precisely the casual understanding of convergence.
> > >
> > > Suppose we choose $T$ such that $T \geq 4 /\epsilon^2$ for some $\epsilon$. It is not mathematically rigorous to claim that $T$ iterations of the algorithm lead the sequence of points to the $\epsilon$-critical point. This is because $\epsilon$ must be an arbitrary number. That is, choosing $T$ for a fixed $\epsilon$ is not a strict guarantee of convergence. Therefore, the output $x_T$ of the algorithm in the previous study you presented is not guaranteed to be $\epsilon$-critical point. So most importantly, this consideration of the behavior of the quantity of interest $g_t$ for a fixed $\epsilon$ or fixed $T$ does not lead to the claim that the output $x_t$ is an $\epsilon$-critical point.
> > >
> > > You are right, this logic may be useful to consider how many iterations are needed to guarantee a certain thresholds. Whether it is standard in the optimization community or not is not for us to judge, but even if it is, we would argue that it should never be treated as a mathematically rigorous convergence analysis. We have added a note to this explanation in Section 1.2 to make our point clearer. The example you provided was so clear that we have included it as part of our argument. Thank you very much.
> > >
> > > ---
> > >
> > > **Reply to Comment 5:** Thanks for pointing this out. In our experiments, Mokhtari's SFW is different from SFWM with fixed momentum that we consider, because Mokhtari's SFW varies momentum according to the schedule proposed in the original paper. We have stated clearly that we only consider fixed momentum and that Mokhtari's SFW is a SFW with a scheduled momentum.
> > >
> > > ---
> > >
> > > **Reply to Comment 6:** You are right, our original manuscript was needlessly complicated. We have unified the convergence metrics and combined all the decreasing learning rate results into the same theorem (revised Theorem 3.3). We have also added new results for decreasing learning rate (I) and increasing batch size (revised Theorem 3.4(i)).
> > >
> > > ---
> > >
> > > **Reply to Comment 7:** We agree with you and have added Adam's tuning details to the main body (Section 3.3).
> > >
> > > ---
> > >
> > > **Reply to Comment 8:** Of course we appreciate your point and have provided a new citation. We were just concerned that this lack of citation would be considered a factor in the rejection.
> > >
> > > ---
> > >
> > > **Reply to Comment 9:** Since our analysis (SFWM and SFW) includes FW, we do not consider Section 4 to be completely separate from Section 3. Regarding the convergence guarantee of AFW, we see from Theorem 3.1 that AFW has a convergence rate of $\mathcal{O}\left(\frac{1}{N_\text{iter}}+\gamma_0 \right)$ if the attacks by AFW are consistently successful, since AFW uses the same learning rate for all iterations. However, it is difficult to guarantee convergence otherwise, and this is true for APGD and ACG as well.
> > >
> > > We were unable to demonstrate an advantage over APGD; however, given that APGD is a state-of-the-art method, we believe that the fact that AFW achieves comparable performance to APGD against robust models still holds value.
> > >
> > > More detailed descriptions of previous studies have been added.
> > >
> > > ---
> > >
> > > **Reply to small errors:** Thank you for pointing out our typographical error. We have corrected it appropriately.

---

> > > > ### Comment · Reviewer_8w4Z · 2025-03-24
> > > >
> > > > ### Comment 2
> > > >
> > > > "It is not mathematically rigorous to claim that iterations of the algorithm lead the sequence of points to the -critical point"
> > > >
> > > > First, the algorithm outputs a single point $x_T$, not a sequence, and it absolutely is mathematically rigorous to claim that the output $x_T$ is an $\epsilon$-critical point. One fixes $\epsilon>0$ before one runs the algorithm. One runs the algorithm for $T$ iterations. After $T$ iterations, one has a point $x_T$ such that $g_T \leq \epsilon$. So $x_T$ is an $\epsilon$-critical point, by definition. This is not a casual understanding of convergence, I am not claiming that any sequence converges to $0$ (and indeed we have just an output point $x_T$, not a sequence, which is an $\epsilon$-critical point).
> > > >
> > > > "You are right, this logic may be useful to consider how many iterations are needed to guarantee a certain thresholds."
> > > >
> > > > Guaranteeing a certain threshold for $g_T$ is exactly what it means to be an $\epsilon$-critical point: $x_T$ is an $\epsilon$-critical point if and only if $g_T \leq \epsilon$.
> > > >
> > > > I want to be clear, I am not claiming there is no difference between showing an algorithm produces a sequence of points whose gaps converge to $0$ and showing an algorithm produces an $\epsilon$-critical point, indeed the former is a stronger result. But, showing an algorithm produces an $\epsilon$-critical point and characterizing how many iterations it takes to do so is not impractical or lacking rigor.
> > > >
> > > > ---
> > > > #### Other comments
> > > >
> > > > There is a new statement added in the beginning of Section 2 as well as a column in Table 1 for boudned gradients,
> > > >
> > > > "Note that, in contrast to many previous studies of SFW in nonconvex optimization (see Table 1), our analysis does not require the assumption of boundedness of the gradient."
> > > >
> > > > I don't understand this statement. You are assuming that the function is continuously differentiable on a compact set; its gradient is necessarily bounded, and so your assumption is even stronger than just a bounded gradient.

---

> > > > > ### Author Response · Authors · 2025-03-25
> > > > >
> > > > > Thank you very much for your prompt reply. Please do not hesitate to reply if you are not satisfied.
> > > > >
> > > > > ---
> > > > > **Reply to Comment 2:** Apparently we misunderstood the definition of epsilon-critical point. Thanks for your point.
> > > > >
> > > > > For $g_T \leq \frac{A}{T\gamma} + B\gamma$, $A=1,B=1$ in your example.
> > > > > However, what happens if $A$ and $B$ are extremely large numbers, such as $10^{10}$?
> > > > >  In general, no one knows what values they will take. In this case, the upper bound $\epsilon$ will also be large, which is not a guarantee of convergence. Or, for sufficiently small $\epsilon$, the required $T$ will be extremely large, so it cannot be said to be a guarantee of convergence. In other words, even if the inequality $g_t<X$ is very loose in this logic, we can call it a convergence analysis if $X<\epsilon$ for some $\epsilon$. This is the kind of looseness that exists in this logic.
> > > > > In contrast, our analysis, which does not use $\gamma_t = 1/\sqrt{T}$, is mathematically guaranteed to converge strictly to zero.
> > > > > We have edited Section 1.2 based on this discussion. Please check it.
> > > > >
> > > > > ---
> > > > >
> > > > > **Reply to other comments:** We agree with you. We have removed the relevant parts. Thank you for pointing this out.

---

> > > > > > ### Comment · Reviewer_8w4Z · 2025-03-25
> > > > > >
> > > > > > ### Comment 2
> > > > > >
> > > > > > Yes, it's true that the constants matter but typically the constants are worse when the step size does not depend on $T$ since you have some partial sum of step sizes involved (this is not a theorem but a general intuition). In the case of Frank-Wolfe algorithms, the important constants (the diameter of the set, the Lipschitz constant, the variance of the stochastic oracle, the functional gap at initialization) have nothing to do with the step size.
> > > > > >
> > > > > > If one were to use your convergence rates (e.g., for decaying step size and increasing batch size as in Theorem 3.4 (ii)) to compute how many iterations are necessary for an $\epsilon$-critical point then they would run into the same problem as far as the constants being big. Having your algorithm produce a theoretical sequence for which the gap converges to $0$ does not mitigate this problem in the slightest.
> > > > > >
> > > > > > "Several previous studies provide convergence analyses of SFW methods for nonconvex optimization, but many analyses do not actually show that the convergence measure is zero for a sufficiently large number of steps."
> > > > > >
> > > > > > This sentence is not meaningful, virtually no analyses show that the convergence measure is $0$ for a sufficiently large number of iterations - this would mean convergence in finite-time and, since most nonconvex problems are NP-hard, this is conjectured to be unreasonable. This also has nothing to do with whether or not the step size depends on $T$. Even your result in Theorem 3.4 (ii) does not show that the gap is $0$ after a large number of iterations - it only gives an estimation of how many iterations you need to run to get an $\epsilon$-critical point.

---

> > > > > > > ### Author Response · Authors · 2025-03-25
> > > > > > >
> > > > > > > Thank you very much for your prompt reply.
> > > > > > >
> > > > > > > ---
> > > > > > >
> > > > > > > **Reply to Comment 2:**
> > > > > > > In the analysis of (Reddi et al. 2016), the step size includes the diameter of the set, the Lipschitz constant, and the function gap at initialization.
> > > > > > >
> > > > > > > Our Theorem 3.4(ii), (iii) has a convergence measure of zero for a sufficiently large number of iterations, since the gap is zero at $T \to \infty$. This is because the step size does not depend on $T$.
> > > > > > >
> > > > > > > ---
> > > > > > >
> > > > > > > Apparently, the reviewer argues that an analysis in which the gap converges to zero is worthless, and we rather consider this to be our contribution, as it is the result that shows true convergence.
> > > > > > > Fortunately, this discussion is disconnected from the soundness of our results, so we are not convinced, but if there is really no value in a result where the gap converges to 0 in $T \to \infty$, and if other reviewers and AEs also want to remove the statement in Section 1.2, we are willing to accept it.

---

> ### Comment · Reviewer_8w4Z · 2025-03-25
>
> That's a fair point, the results of Reddi et al do require these constants, but it's not the case for instance in Pethick et al, which is more relevant to this context anyway since their results are for SFWM.
>
> >Apparently, the reviewer argues that an analysis in which the gap converges to zero is worthless
>
> I never said such a thing, please don't mischaracterize me. I agree your results show convergence of the gap to $0$ and this is slightly stronger than just showing the algorithm outputs an $\epsilon$-critical point. I disagree that this has meaningful impact on practice and I disagree with the mathematical pertinence of the claim in the paper that I already quoted,
>
> "Several previous studies provide convergence analyses of SFW methods for nonconvex optimization, but many analyses do not actually show that the convergence measure is zero for a sufficiently large number of steps."
>
> This is not true for your algorithm either, your analysis does not show that the gap is $0$ for any finite number of steps.

---

> > ### Author Response · Authors · 2025-03-25
> >
> > We greatly appreciate your prompt reply.
> >
> > ---
> >
> > Our explanation may indeed be confusing. We have revised it to make the meaning clearer.
> >
> > We sincerely appreciate the careful and detailed explanations.

---

### Review · Reviewer_7TWE · 2025-03-01

**Summary Of Contributions:**

In the submitted paper authors investigate Stochastic Frank-Wolfe algorithm with momentum and increasing batch size for nonconvex problems. They derive the convergence rates, depending on various method's regimes. Furthermore, the authors show that the proposed algorithm with adaptive step selection is competitive to APGD for white-box adversarial attacks.

**Audience:**

Yes

**Broader Impact Concerns:**

There are no potential societal consequences, that should be specifically highlighted.

**Claims And Evidence:**

Yes

**Requested Changes:**

$\bullet$ Since one of the advantages of the Frank-Wolfe algorithm is the simplicity of iteration on a bounded set, consider adversarial attacks with various $l_p$ constraints, not only $l_{inf}$,

$\bullet$ It might be thoughtful to compare consumed resources and time in the experiments, since there might be an optimal batch size.

**Strengths And Weaknesses:**

Strengths:

$\bullet$ Comprehensive and clear analysis of existing methods,

$\bullet$ Usage of parameter-free decaying learning rate and increasing batch size,

$\bullet$ Decent performance in the adversarial attacks on classification models.

Weaknesses:

$\bullet$ Though, the convergence rate for the 3-rd option of the decaying learning rate with increasing batch size is claimed to be $\mathcal{O}\left(\frac{1}{T}\right)$, in Theorem 3.5 one can notice $\underline{\gamma}$ and $\gamma_{inf}$, that depend on $T$ and are not thoroughly examined,

$\bullet$ Besides not mentioning learning rates and batch sizes in the adversarial attacks' description, it is unclear how much resources or time was spent on APGD and ASFW. Projection-free methods might be more effective, than projection-based,

$\bullet$ Technical inaccuracies, such as $\mathbb{E}\left[f(\theta_0)\right] - \mathbb{E}\left[f(\theta^*)\right]$ (both $\theta_0$ and $\theta^*$ are not random) or invalid reference on $A.1$ on page 20.

---

> ### Author Response · Authors · 2025-03-14
>
> Thank you for your thoughtful review. Please find the revised paper and our responses below. If you have any further concerns, please let us know.
>
> ---
>
> **Weakness 1:** Though, the convergence rate for the 3-rd option of the decaying learning rate with increasing batch size is claimed to be $\mathcal{O}(1/T)$, in Theorem 3.5 one can notice $\underline{\gamma}$ and $\gamma_\infty$, that depend on $T$ and are not thoroughly examined.
>
> **Reply:** We agree with you. To address this point, we added Lemma A.5 and eliminated $\gamma_\infty$ from the theorem. We have also explicitly defined a lower bound of the learning rate independent of $T$, the $\underline{\gamma}$. These corrections should address the reviewers' concerns. If you still have concerns, please let us know.
>
> ---
>
> **Weakness 2:** Besides not mentioning learning rates and batch sizes in the adversarial attacks' description, it is unclear how much resources or time was spent on APGD and ASFW. Projection-free methods might be more effective, than projection-based,
>
> **Reply:** We agree with you. Our manuscript lacked an explanation of the learning rate, so we have added that explanation in Section 4.5. Also, neither APGD nor our proposed algorithm performs stochastic operations. For practical purposes, both algorithms require a batch size $b$ as an argument, but this simply means that $b$ attacks are performed in order, and there is no random element in it (See our anonymous GitHub). Therefore, our manuscript contains an error, and our algorithm should naturally be called Auto-Frank Wolfe (AFW), not Auto-Stochasitic Frank Wolfe (ASFW). We apologize for any confusion this may have caused.
>
> In addition, the computational complexity and time for APGD and AFW are comparable, and unfortunately we could not find any advantage of AFW in this respect either. We have explicitly added a description of this as well in section 4.5.
>
> ---
>
> **Weakness 3:** Technical inaccuracies, such as $\mathbb{E}\left[f(\theta_0)\right] - \mathbb{E}\left[f(\theta^*)\right]$ (both $\theta_0$ and $\theta^\star$ are not random) or invalid reference on A.1 on page 20.
>
> **Reply:** We removed the expected values of $f(\theta_0)$ and $f(\theta^\star)$. Thanks for your helpful comments.
>
> ---
>
> **Requested Change 1:** Since one of the advantages of the Frank-Wolfe algorithm is the simplicity of iteration on a bounded set, consider adversarial attacks with various $l_p$ constraints, not only $l_\text{inf}$.
>
> **Reply:** To address your plausible point, we have added new results of attacks by APGD and AFW when the $L_1$ constraint is imposed on adversarial perturbations (see Table 4 in revised manuscript). Thanks for your valuable comments.
>
> ---
>
> **Requested Change 2:** It might be thoughtful to compare consumed resources and time in the experiments, since there might be an optimal batch size.
>
> **Reply:** We apologize for any inconvenience caused by our mistake. Our proposed algorithm (AFW) is not stochastic and always works in full batch. See also reply to Weakness 2.

---

> > ### Author Response · Authors · 2025-03-27
> >
> > Dear Reviewer 7TWE,
> >
> > We sincerely appreciate your time and feedback. We have carefully addressed the reviewers' comments and revised the manuscript accordingly. If you have any remaining questions or concerns, please do not hesitate to let us know. We would be happy to address them. Thank you for your time and consideration.

---

### Decision · Action_Editor_ZRLF · 2025-04-13

**Recommendation:** Reject

**Comment:**

The paper provides several new results on the convergence of the Stochastic Frank-Wolfe method with momentum for non-convex constrained optimization. However, there are certain aspects of the paper requiring major revision, as mentioned by all the reviewers.

In particular, as Reviewer 7TWE mentioned in the final recommendation, the dependence on $\overline{\gamma}$ is problematic in Theorem 3.4: since $\underline{\gamma} = \eta^{P-1} \gamma$, it can be exponentially small. In this case, Theorem 3.4 (iii) does not imply $O(1/T)$ convergence. Moreover, the reviewers mentioned that the analysis requires the usage of large/increasing batchsizes, which is not practical. I also went through the back-and-forth discussion between the authors and Reviewer 8w4Z about the horizon-dependent stepsizes. I believe the authors can simply add that they are interested in horizon-independent stepsizes -- this will make the discussion in Section 1.2 much clearer.

**Audience:**

Yes, the paper would be of interest to the optimization community at TMLR.

**Claims And Evidence:**

Two out of three reviewers answered "Yes" to this question.

**Resubmission Of Major Revision:**

The authors may consider submitting a major revision at a later time.